# A framework for multiplex imaging optimization and reproducible analysis

Jennifer Eng [1,2], Elmar Bucher [1], Zhi Hu[1], Ting Zheng[3], Summer L. Gibbs[1,2], Koei Chin [1,2 ✉] & Joe W. Gray [1,2 ✉]

Multiplex imaging technologies are increasingly used for single-cell phenotyping and spatial characterization of tissues; however, transparent methods are needed for comparing the performance of platforms, protocols and analytical pipelines. We developed a python software, mplexable, for reproducible image processing and utilize Jupyter notebooks to share our optimization of signal removal, antibody specificity, background correction and batch normalization of the multiplex imaging with a focus on cyclic immunofluorescence (CyCIF). Our work both improves the CyCIF methodology and provides a framework for multiplexed image analytics that can be easily shared and reproduced.

[1] Department of Biomedical Engineering, School of Medicine, Oregon Health and Science University, Portland, OR 97239, USA. [2] Knight Cancer Institute, School of Medicine, Oregon Health and Science University, Portland, OR 97239, USA. [3] Cancer Early Detection Advanced Research Center, School of Medicine, Oregon Health and Science University, Portland, OR 97239, USA. ✉email: ChinKoei@ohsu.edu; grayjo@ohsu.edu

Multiplex imaging methods enable the quantification of numerous proteins in tissues while retaining spatial and morphological information. In many contexts, dozens of targets must be labeled in an intact tissue section to identify key cell types and quantify their spatial relationships. Multiplex imaging technologies utilize various strategies to overcome the limitation of conventional immunofluorescence (IF) protocols that label a few markers per section. Five and seven-plex immunohistochemistry (IHC) has been achieved by fluorophore-conjugated tyramide deposited on the tissue[1–3]. Twelve to 29-plex immunohistochemistry (IHC) is enabled with alcohol-soluble peroxidase substrate 3-amino-9-ethylcarbazole (AEC) detection combined with antibody stripping[4,5] and 40-plex IF can be achieved with antibody elution in iterative indirect immunofluorescence imaging (4i)[6] and multiple inter-active labeling by antibody neodeposition (MILAN)[7,8]. Direct immunofluorescence using fluorophore-conjugated primary antibodies and chemical inactivation of fluorescent dyes enables detection of over 50 protein targets in a single tissue section, in cyclic immunofluorescence (CyCIF)[9–11], NeoGe-nomics' MultiOmyx platform[12], and iterative bleaching extends multiplexity (IBEX)[13]. Similarly, multi-epitope-ligand carto-graphy (MELC), employs photo-inactivation of fluorescently labeled antibodies[14,15]. Furthermore, fluorophore-conjugated DNA barcodes (i.e., oligonucleotides) facilitate multiplexing in co-detection by indexing (Akoya's CODEX)[16], Ultivue's InSituPlex[17], Immuo-SABER[18], and Ab-oligo cyCIF[19]. Imaging mass cytometry[20] and multiplex ion beam imaging[21] can detect over 40 antibodies conjugated to metal reporters by time-of-flight mass spectrometry. The Nanostring GeoMx[22] platform can detect >100 protein targets conjugated to oligonucleotide barcodes, although the data are not images, but spatially registered counts of released oligos[23]. All of these methods rely on specific labeling of proteins with antibodies, a process that must be validated by studies that establish specificity, sensitivity, and reproducibility. Ideally this is accomplished via transparent analyses using well-documented, open-source image analysis pipelines.

Herein, we quantitatively assess antibody labeling in one multiplex imaging platform, CyCIF[9–11], while varying antibody application strategies, fluorescence signal and tissue auto-fluorescence removal methods, and experimental batches. This resulted in several strategies to improve CyCIF, including opti-mization of aspects of chemical bleaching, autofluorescence cor-rection, antibody application order, and batch normalization. Herein, we build on protocol optimization efforts[10,24] and share our image processing pipeline software, the free and open-source python library, mplexable (Fig. 1a, b). Finally, we introduce a framework for reproducible image analysis and visualization. Our analytics comprise linked and executable code and data that enabled all figures and analytical results in our paper to be fully reproduced from the accompanying image data (https://www.synapse.org/#!Synapse:syn23644107), processed data, and code (https://github.com/engjen/cycIF_Validation). We used Jupyter notebooks to link code, data, metadata and the computational environment in a machine- and human-readable document (Fig. 1c and Table 1). Although we focused on validation of a single platform, our tools for image analysis facilitate quantitative and reproducible characterization of tissues by any multiplex imaging platform.

## Results and discussion
Adopting the method of Lin et al.[9], we stained formalin-fixed, paraffin-embedded (FFPE) tissues using a cyclic process in which four proteins per cycle were labeled using a direct immuno-fluorescence (IF) protocol, imaged, and quenched of fluorescence

signal (Fig. 1a, https://doi.org/10.17504/protocols.io.23vggn6). We typically labeled 40–60 proteins per slide before tissue and staining quality degraded (quantified in Fig. 2e and Supplemen-tary Fig. 7, respectively). We developed free and open-source tools to perform quality control on images and metadata and automate image processing from registration through single-cell segmentation and feature extraction (Fig. 1b, https://pypi.org/project/mplexable/). Finally, we shared all analytics used to pro-duce figures and conclusions as linked and executable code and data (Fig. 1c). Jupyter notebooks serve as a human and machine-readable record that described the data, image analysis steps, and computational environment, both enabling replication of our findings and linking data and metadata to facilitate use by others.

**Comparison to a standard IF.** We benchmarked our CyCIF protocol against a direct IF protocol. CyCIF staining in normal and malignant breast tissue was compared to standard direct IF on adjacent tissue sections. Antibodies and staining conditions were selected that produced visually similar staining patterns in both CyCIF and standard IF staining (Fig. 2a, Notebook 1-1, NB1-1). For quantification, we set an intensity threshold for each marker, with all pixels above the set threshold considered positive. All pixels at least 10 μm away from positive staining were considered background pixels (the 10 μm gap was to exclude any influence of lateral bleed through around positive pixels; Fig. 2b and Supple-mentary Fig. 1, NB2-1). If a marker was negative in a given tissue section, it was not analyzed in that tissue. Although the positive and background intensity varied between conditions, presumably due to the imprecision introduced by manual pipetting, the rela-tive signal-to-background ratio (SBR, see methods) of standard IF compared to CyCIF was near one (relative SBR: mean = 0.96, standard error of the mean [SEM] = 0.22) and highly correlated (Pearson $R = 0.89$, $p = 1.6e-9$, Fig. 2c and Supplementary Fig. 1, NB2-2). We assessed specificity between conditions by segmenting single cells and counting positive cells with a mean intensity above the set threshold. We found that the percentage of positive cells of standard IF relative to CyCIF was also near one (relative % positive: mean = 0.96, SEM = 0.18) and the fraction positive was highly correlated (Pearson $R = 0.99$, $p = 5.8e-24$, Fig. 2c and Supplementary Fig. 1c, NB2-2). We concluded that the SBR and specificity of antibody staining in the first five cycles of CyCIF is similar to a standard IF, the cyclic process does not excessively impact tissue staining, and CyCIF offers the advantage of detecting increased marker combinations while utilizing a single slide.

**Tissue loss characterization.** The CyCIF process resulted in increased tissue loss compared to a standard IF (Supplementary Fig. 1a, NB2-1). In order to characterize the numbers and types of cells and tissues that were susceptible to tissue loss during CyCIF, three adjacent sections of a 72-core tissue microarray (TMA) containing normal and malignant tissues were repeatedly quen-ched and imaged (Fig. 2d). Slow but steady tissue loss was observed over ten rounds of CyCIF, with 95% of cells remaining after ten rounds (Fig. 2e). We found that the 18 normal tissues suffered more tissue loss than the 52 malignant and two benign tissues (Fig. 2e), and the difference in fraction of cells remaining after ten rounds was significant (Fig. 2f). We found no difference in the number of cells lost in tissues separated by stage or grade, or by cells separated by their nuclear size and shape (which correlate with epithelial, immune, endothelial, or mesenchymal cell types, Fig. 2f–h). While all malignant tissues were obtained from autopsy, normal tissues obtained from sur-gical resections vs. autopsy showed a trend towards reduced

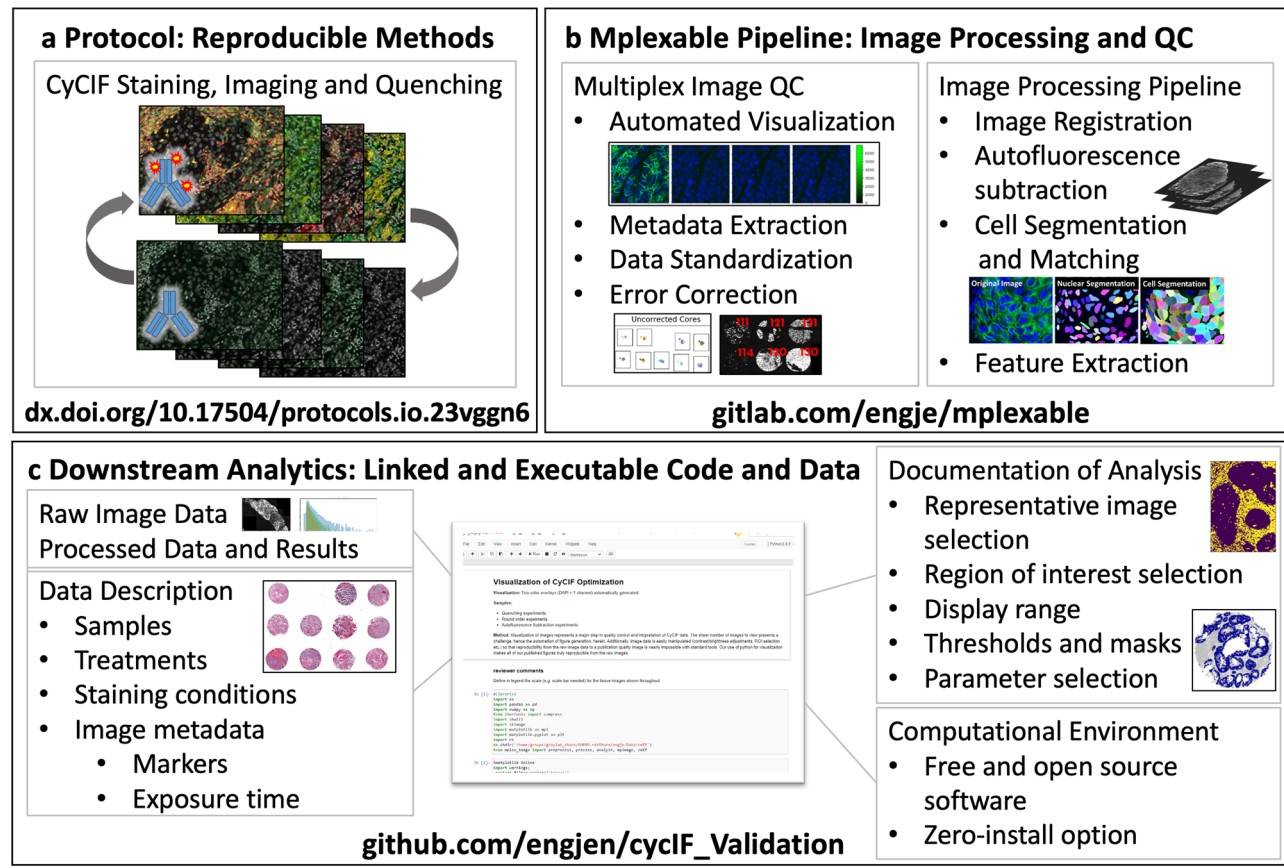

**Fig. 1 Reproducible generation, processing, and analysis of multiplex images. a** CyCIF data is generated with a reproducible protocol (dx.doi.org/10.17504/protocols.io.3xfgpjn). **b** Image processing pipeline integrates quality control and workflow management with in-house Python software, mplexable, facilitating data quality, scaling, and reproducibility. **c** Analysis, image visualization, and figure generation are fully reproducible with shared linked and executable code and data. Raw and processed data, metadata, and analysis are documented in Jupyter notebooks running our free and open-source software that provides a framework for reproducible image analysis.

**Table 1 Reproducible image analytics.**

| Name | Links to analysis notebooks<br>url |
|---|---|
| NB1-1 | https://github.com/engjen/cycIF_Validation/blob/master/Multicolor_Image_Visualization.ipynb |
| NB2-1 | https://github.com/engjen/cycIF_Validation/blob/master/Extended_single_vs_cyclic.ipynb |
| NB2-2 | https://github.com/engjen/cycIF_Validation/blob/master/SinglevsCyclic_44290.ipynb |
| NB3-1 | https://github.com/engjen/cycIF_Validation/blob/master/Quenching_analysis.ipynb |
| NB3-3 | https://github.com/engjen/cycIF_Validation/blob/master/Image_Analysis_Visualization.ipynb |
| NB3-4 | https://github.com/engjen/cycIF_Validation/blob/master/DoubleApplication_K157.ipynb |
| NB3-5 | https://github.com/engjen/cycIF_Validation/blob/master/OrderOptimization_K154vsK175.ipynb |
| NB3-6 | https://github.com/engjen/cycIF_Validation/blob/master/Macsima_clustering.ipynb |
| NB4-1 | https://github.com/engjen/cycIF_Validation/blob/master/Quenching_Single_Cell.ipynb |
| NB5-1 | https://github.com/engjen/cycIF_Validation/blob/master/Extended_Reproducibility_3TMA_Tissue.ipynb |
| NB5-2 | https://github.com/engjen/cycIF_Validation/blob/master/TMAReplicates_analysis.ipynb |
| NB5-3 | https://github.com/engjen/cycIF_Validation/blob/master/Normalization_testing_tissue.ipynb |
| NB5-4 | https://github.com/engjen/cycIF_Validation/blob/master/Normalization_testing_HER2-N75.ipynb |
| NB5-5 | https://github.com/engjen/cycIF_Validation/blob/master/Normalization_testing.ipynb |
| NB5-6 | https://github.com/engjen/cycIF_Validation/blob/master/kBET.ipynb |
| NB5-7 | https://github.com/engjen/cycIF_Validation/blob/master/RestoreNorm_scale.ipynb |

All figures and results can be fully reproduced with the accompanying Jupyter notebooks and data provided here: https://github.com/engjen/cycIF_Validation, https://doi.org/10.5281/zenodo.6049278.

tissue loss (Mann–Whitney $U$, $p = 0.083$). This could reflect tissue processing variation, as smaller tissue size and longer time in fixation have been shown to reduce section detachment in immunohistochemistry[25]. There were significant differences in tissue loss in normal tissues from different organs and malignant tissues with different pathologies (Supplementary Fig. 3). Our results suggest that tissue loss varied more between different tissues than between individual cell types within a given tissue.

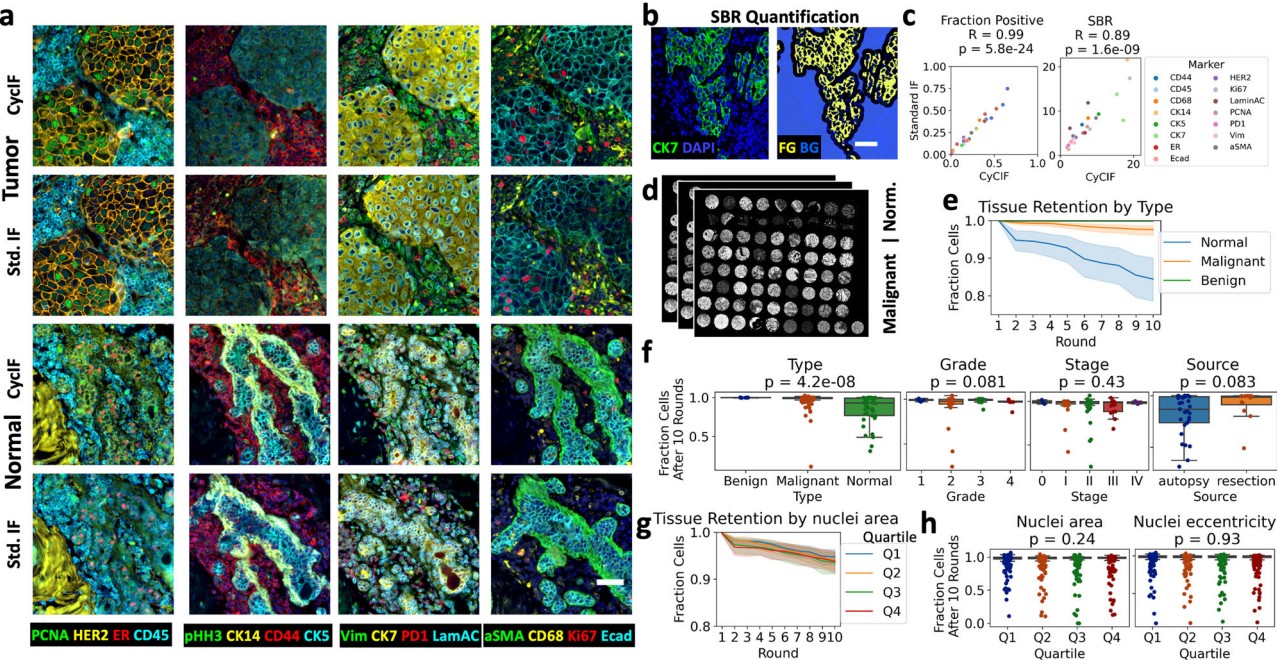

**Fig. 2 CyclF signal to background and tissue loss.** Comparison of CyCIF staining to a standard IF protocol, in adjacent sections of HER2 + tumor and normal breast tissue. **a** Multicolor visualizations are produced with our python code. **b** Positive pixels or foreground (FG), were determined by thresholding and negative pixels or background (BG), were determined by selecting regions 10 μm away from positive regions. The signal-to-background ratio (SBR) is the mean intensity of FG/mean intensity of BG. **c** Positive cell counts were determined by applying the thresholds from (**b**) to single-cell mean intensities of segmented cells. Pearson correlation ($R$), was calculated for standard IF versus CyCIF fraction positive in tissue and SBR, $n = 26$; $R$ and $p$ value given in figure title. **d** Three adjacent sections of a 72-core tissue microarray (TMA) containing normal and malignant tissues were repeatedly quenched to assess tissue loss during CyCIF. **e** Tissue retention after each round of quenching. Error is a 95% confidence interval of tissue retention at each round for normal, benign, and malignant tissues, $n = 54$, 6, and 156, respectively (18 normal, 2 benign, and 52 malignant tissues × 3 replicate TMAs). **f** Fraction of cells remaining after ten rounds of CyCIF, separated by type, stage, grade, and source. **g** Tissue retention by nuclear area quartile; error is 95% confidence interval, $n = 216$ cores. **h** Fraction of cells remaining after ten rounds of CyCIF, separated by nuclear area quartile and nuclear eccentricity quartile. In f, h, Kruskal–Wallis $H$-test was used to assess differences in tissue retention, $n = 216$ (72 cores × 3 replicate TMAs), except Source, which includes only normal tissue, $n = 41$ cores. $p$ value shown in figure title. In **a**, **b**, scale bar = 50 μm.

**Quenching condition assessment**. Most tissue loss was observed during the quenching step, necessitating the development of gentle yet effective conditions for complete signal removal. CyCIF fluorescence signal removal was accomplished by chemical quenching of fluorescent dyes using hydrogen peroxide ($H_2O_2$). Only certain dyes, including Alexa Fluor (AF)−488, AF555, AF647, and AF750, were quenchable using this method. Other dyes, including AF546 and fluorescein, were resistant to quenching, as previously reported[24]. Increasing $H_2O_2$ concentration from 3 to 4.5% or 6% in 20 mM sodium hydroxide did not improve the quenching rate, and additional time in $H_2O_2$ also failed to completely eliminate strong signal (Fig. 3a, b, d and Supplementary Figs. 4, 5 NB3-1). Gentle heating with an incandescent light placed ~4 inches above the sample during quenching (see methods) appeared to increase the rate of $H_2O_2$ oxidation (estimated by heat generation, Supplementary Fig. 5c), resulting in complete signal removal (Fig. 3c, d and Supplementary Fig. 5a, b, NB3-1). Quantification of the unstained negative controls in all quenching conditions showed that mean tissue autofluorescence decreased during the first quench (AF488: mean 22% decrease, SEM = 2.8%; AF555: mean 24% decrease, SEM = 2%, AF647: mean 5% decrease, SEM = 1.4%, $n = 3$), a decrease not seen in an unquenched tissue that was repeatedly imaged (Supplementary Fig. 5e, f, NB3-1). Therefore, we standardized quenching using 3% $H_2O_2$ for 30 min with incandescent light and added an additional round of quenching (i.e., pre-quenching before CyCIF staining) similar to others[10], to reduce overall tissue autofluorescence by roughly 25% (Supplementary Fig. 5e).

**Antibody order evaluation and improvement**. Finally, we tested if the order of antibody application in the CyCIF panel influenced antibody sensitivity and specificity. We applied an 11-round, 44-antibody panel twice to the same TMA, for a total of 22 rounds (Supplementary Table 1). We visually and quantitatively compared the antibody staining pattern and single-cell mean intensities of the images acquired after the first application of each antibody to the second (Supplementary Fig. 6, NB3-3). Of the 44 antibodies evaluated, 86% had a high Pearson correlation (>0.8) between the first and second applications, but 81% had a lower dynamic range and SBR on the second application (Supplementary Fig. 7, NB3-4). Effects of double-application were antibody and epitope-specific, as 28% of antibodies showed a similar dynamic range (<50% change) on the second application, and 62% still had an estimated SBR >1.5 on the second application, versus 78% on the first (Supplementary Fig. 7, NB3-4). Non-specific nuclear staining was observed for several antibodies conjugated to the AF750 fluorophore, and unexpectedly, it diminished on the second application (Supplementary Fig. 6c, NB3-4). We tested several potential variables impacting stain quality in later rounds, including nonspecific IgG interactions, nonspecific fluorophore interactions, specific IgG interactions, and quenching effects. Out of all of these, the only condition that apparently reduced nonspecific nuclear background was additional quenching (Supplementary Fig. 8). Thus, we designed an improved panel order which positioned antibody-conjugates tending to have nonspecific staining in a later round (Fig. 3e), as well as addressed challenges of autofluorescence, channel bleed

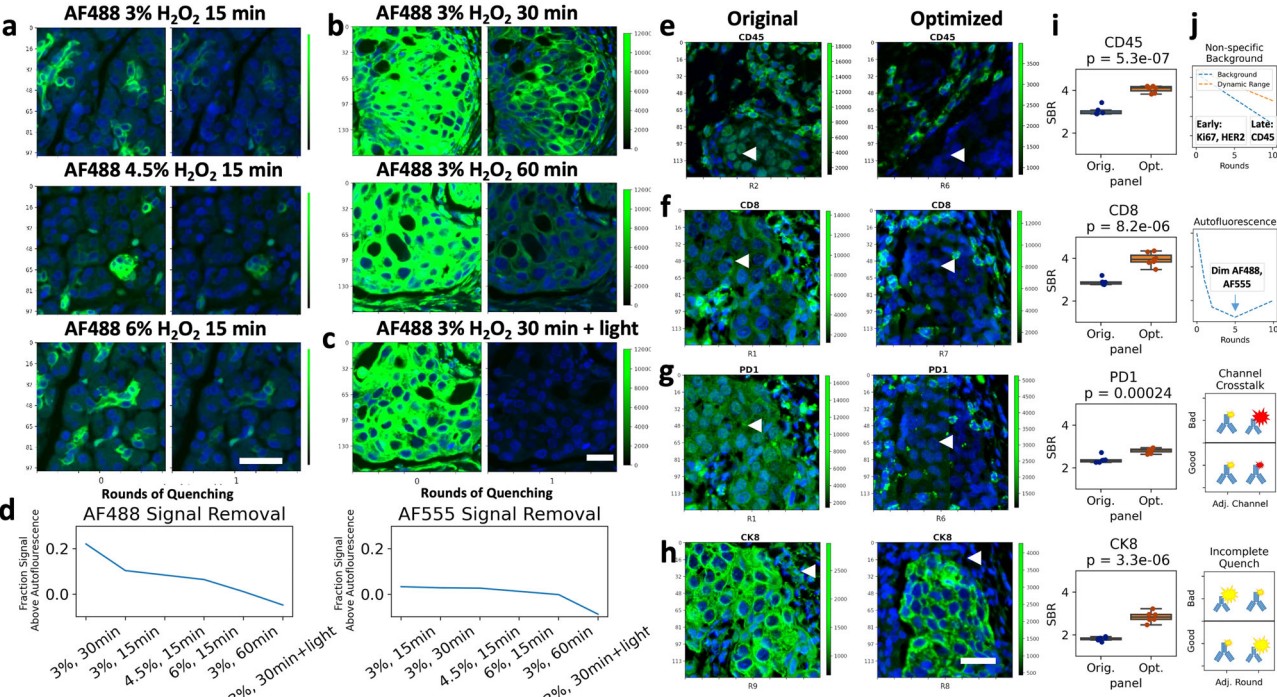

**Fig. 3 CyCIF optimization: quenching and panel order. a–d** Quenching optimization. a Normal pancreas was stained with CK7-AF488 and imaged before and after 15 min of quenching in 3% (top row), 4.5% (middle row), or 6% (bottom row) H₂O₂. **b** Breast cancer tissue stained with CK7-AF488 and Vimentin-AF488 imaged before and after 30 (top) and 60 min of quenching (bottom). **c** Tissue stained as in (**b**) and imaged before and after 30-min quenching under an incandescent light source. **d** Mean AF488 (left) or AF555 (right) fluorescence intensity in tissue area of stained tissue relative to a blank autofluorescence control after one round of quenching with conditions shown in **a–c**. **e–h** Panel order optimization, with representative images. **i** Signal-to-background ratio (SBR) quantification was done by applying a threshold to find positive pixels, and manually selecting areas of nonspecific background, e.g., tumor nests in **e–g** and stromal areas in **h**, n = 6 areas per tissue. A t-test was used to assess significance; p value is shown in the figure title. **e** Nonspecific nuclear staining in the AF750 channel is improved by moving the antibody to a later round. CD45-AF750 showed higher SBR in round (R) 6 than in R2. **f** Autofluorescence in the AF555 or AF488 channel is mitigated by moving the antibody from R1 to a later round. CD8-AF555 had higher SBR in R7 than R1. **g** Channel bleed through is mitigated by pairing two bright or two dim antibodies in adjacent channels, not a bright with a dim. PD1-AF647 shows bleed through from bright CK19-AF750 in R1, but not from CD45-AF750 in R6. **h** Incomplete quenching is mitigated by moving markers resistant to quenching to later in the panel. Vimentin-AF488 resists quenching and is moved after cytokeratin staining, rather than before. **j** Schematic of panel optimizations addressing background and dynamic range, autofluorescence, channel bleed through, and incomplete quenching. **a–c** and **e–h**. Blue = DAPI, Green = Stain, green color bar = 16-bit grayscale intensity, Y-axis scale is in micrometers, scale bar = 30 μm.

through and incomplete quenching (Fig. 3f–j and Supplementary Table 2). We compared the original and optimized orders on near-adjacent slides cut from a TMA containing three tissues with positive staining for each marker. SBR quantification was completed by applying a threshold to each marker to find positive pixels, and manually selecting areas of nonspecific background (e.g., tumor nests for immune markers and stromal areas for tumor markers). Our new panel order significantly increased SBR (Fig. 3i, NB3-5). In earlier rounds, we prioritized antibodies to antigens with varied expression levels such as human epidermal growth factor receptor 2 (HER2), E-cadherin, and cytokeratins, which generally lose their dynamic range in later staining rounds. We placed immune cell markers in later rounds, as they have a binary (i.e., on/off) expression pattern and sometimes showed better SBR in later rounds, due to decreased background (Fig. 3e, f, j). Markers expressed in the same cell type and/or subcellular location were placed in non-adjacent channels and rounds to allow detection artifacts from channel bleed through and any incomplete quenching. Bleed through was evident for weakly staining and strongly staining antibodies in adjacent channels (e.g., PD1-AF647 and CK19-AF750, Fig. 3g, j), but was minimized by avoiding such combinations in the same round. Bleed through was not specific to our selected fluorophores and filter sets, as we observed FITC to PE bleed through in another multiplex imaging platform, which could also be mitigated by

avoiding strongly staining and weakly staining antibodies in adjacent channels (Supplementary Fig. 8, NB3-6).

**Autofluorescence dynamics and corrections**. Although pre-quenching reduced overall tissue autofluorescence by ~25% (Supplementary Fig. 5e), we sought to develop a method to computationally subtract remaining autofluorescence, which, in some tissue structures, was brighter than the biomarker staining. First, we characterized single-cell autofluorescence dynamics over rounds of CyCIF by quantifying single-cell AF488 intensity in three unstained normal pancreas tissues that had been imaged and quenched repeatedly. We segmented the nuclei based on DAPI staining and expanded each nucleus by five pixels to capture the cytoplasm. Tissues were batch normalized and unsupervised clustering revealed groups of cells with comparable autofluorescence profiles over the experiment that localized to similar regions in each tissue (Supplementary Fig. 9, NB4-1). Overlaying the cells in each cluster on the tissue revealed cells in cluster seven comprised tissue structures with bright auto-fluorescence that covered a wide spectrum from DAPI to AF488 channels (Fig. 4a). Quantification of mean cluster intensity over rounds of quenching showed that while other clusters' intensities stayed constant, the cluster seven intensity dropped linearly between one and four rounds of quenching (Fig. 4c, NB3-1, NB4-

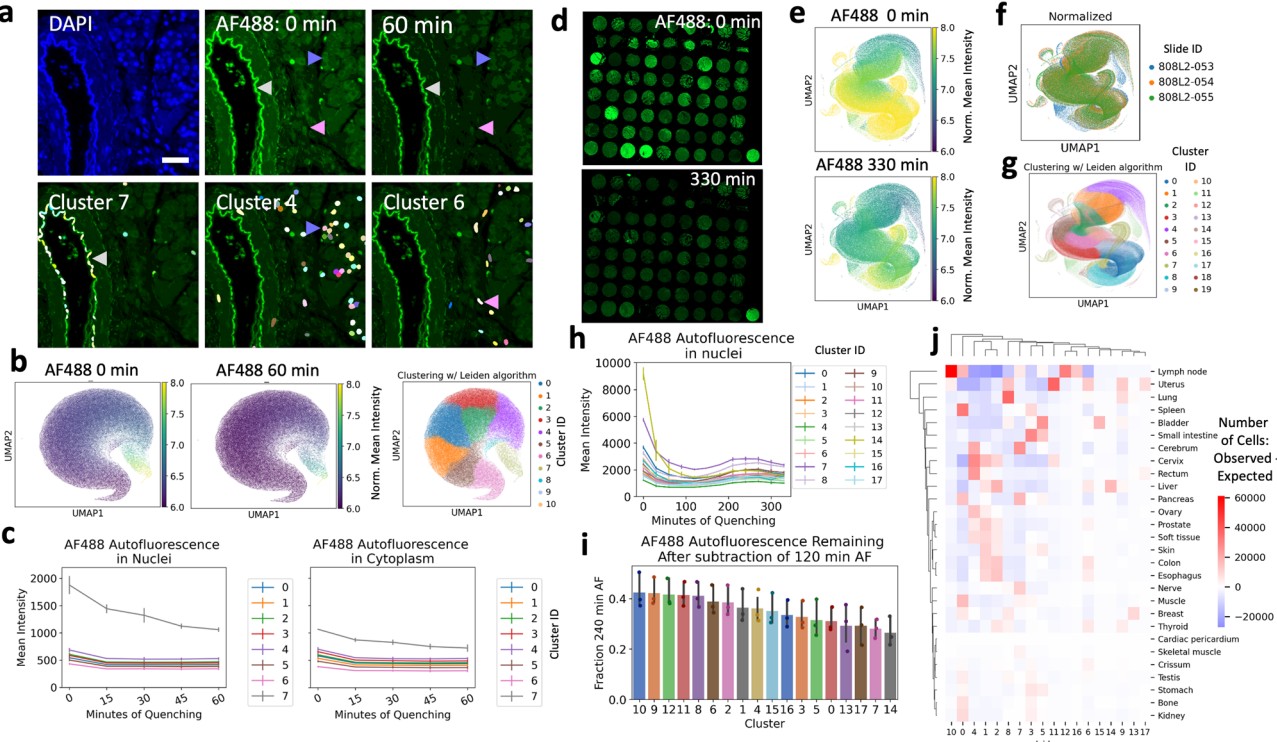

**Fig. 4 Autofluorescence characterization in the CyCIF protocol. a** Top: DAPI nuclear staining and autofluorescence in the AF488 channel before and after 60 min of quenching. Arrowheads show areas of bright (gray), medium (purple), and dim (pink) autofluorescence. Bottom: Colored nuclei of clusters 7 (left), 4 (middle), and 6 (right) on the autofluorescence image, representing cells with bright (gray arrowhead), medium (purple arrowhead), and dim (pink arrowhead) autofluorescence, respectively. Scale bar = 30 μm. **b** UMAP projection of single cells based on autofluorescence, colored by AF488 intensity after 0 or 60 min of quenching (left and middle); Umap colored by unsupervised clustering results of the Leiden algorithm (right). **c** Mean AF488 intensity of cells in each Leiden cluster over rounds of quenching; note bright (gray line) cells quench differently than medium (purple line) and dim (pink line) cells. A similar trend in nuclear (left) and cytoplasmic (right) autofluorescence, although nuclear is brighter. **d** Autofluorescence analysis as in (**a**–**c**) was repeated on a 72-core tissue microarray (TMA). AF488 autofluorescence was shown after 0 min (top) and 330 min or quenching (bottom). **e** UMAP projection of single cells based on autofluorescence, colored by AF488 intensity after 0 (top) or 330 min of quenching (bottom). **f** UMAP colored by batch. Three adjacent sections from TMA were repeatedly quenched and normalized by batch for analysis. **g** UMAP colored by unsupervised clustering results of the Leiden algorithm. **h** Mean AF488 intensity of cells in each Leiden cluster over rounds of quenching. Overall, the minimum intensity was observed at 120 min and maximum intensity at 240 min. **i** Subtracting the minimum intensity autofluorescence will avoid over-subtraction while still removing 60−70% of autofluorescence at 240 min. **j** Heatmap showing an observed number of cells per Leiden cluster —expected number of cells per cluster, by tissue, y-axis, illustrating different trends by tissue. The brightest autofluorescence, cluster 14 (yellow line, **h**), originates from liver tissue. Cluster 10, showing a greatest fraction of autofluorescence remaining after subtraction (blue bar, **i**), originates primarily in a lymph node. **c**, **h**, and **i**. Error bars = standard error of the mean, n = 3 slides.

1). To confirm these results, we collected two additional datasets, three adjacent sections from a 72-core TMA containing normal and tumor tissues (Fig. 4d) and three adjacent sections from an 11-core TMA containing HER2 + breast cancer tissues. The two groups of slides were quenched for a total of ten rounds, 330 min and six rounds, 210 min, respectively. We segmented cells to obtain single-cell autofluorescence values, batch normalized and clustered as described for the pancreas tissue (Fig. 4e–g and Supplementary Fig. 10). Although the AF488 autofluorescence initially declined in these datasets, as had been observed in the pancreas, surprisingly it increased in later rounds, globally across all clusters (Fig. 4h). We did observe that the autofluorescence clusters correlated with one or more tissue types and pathologies (Fig. 4j and Supplementary Fig. 11), but they all showed the same trend, with a minimum intensity at three or four rounds of quenching (after pre-quenching, Fig. 4h and Supplementary Fig. 12). We saw similar trends in AF555 autofluorescence (Supplementary Figs. 11, 12). Therefore, to computationally remove autofluorescence without over-subtracting, we recommend collecting baseline autofluorescence images after quenching at round three or four, and subtracting these, scaled by exposure time, from AF488, AF555, and AF647 channels. Subtraction from the AF750 channel appeared unnecessary, given its minimal autofluorescence. This procedure will remove an additional 60–70% of autofluorescence from the brightest rounds (Fig. 4i and Supplementary Figs. 11, 12). Since different tissues and other multiplex imaging platforms may exhibit different autofluorescence dynamics, we implemented both the baseline algorithm and a scaled algorithm, which assumes linear increase or decrease in autofluorescence, for autofluorescence subtraction in mplexable. We found that the scaled algorithm, by linearly interpolating autofluorescence between background images collected at the beginning and end of staining, did reduce false positives for some markers in our panel (Supplementary Fig. 11). Since the brightest areas of autofluorescence showed linear decrease across the first few rounds of CyCIF, the scaled algorithm was most advantageous in tissues with strong autofluorescence that are stained with only a few rounds of CyCIF. On the other hand, the baseline algorithm is simple, requiring collection of just one background image, and can remove the majority of autofluorescence background without over-subtraction in various tissue and experimental contexts.

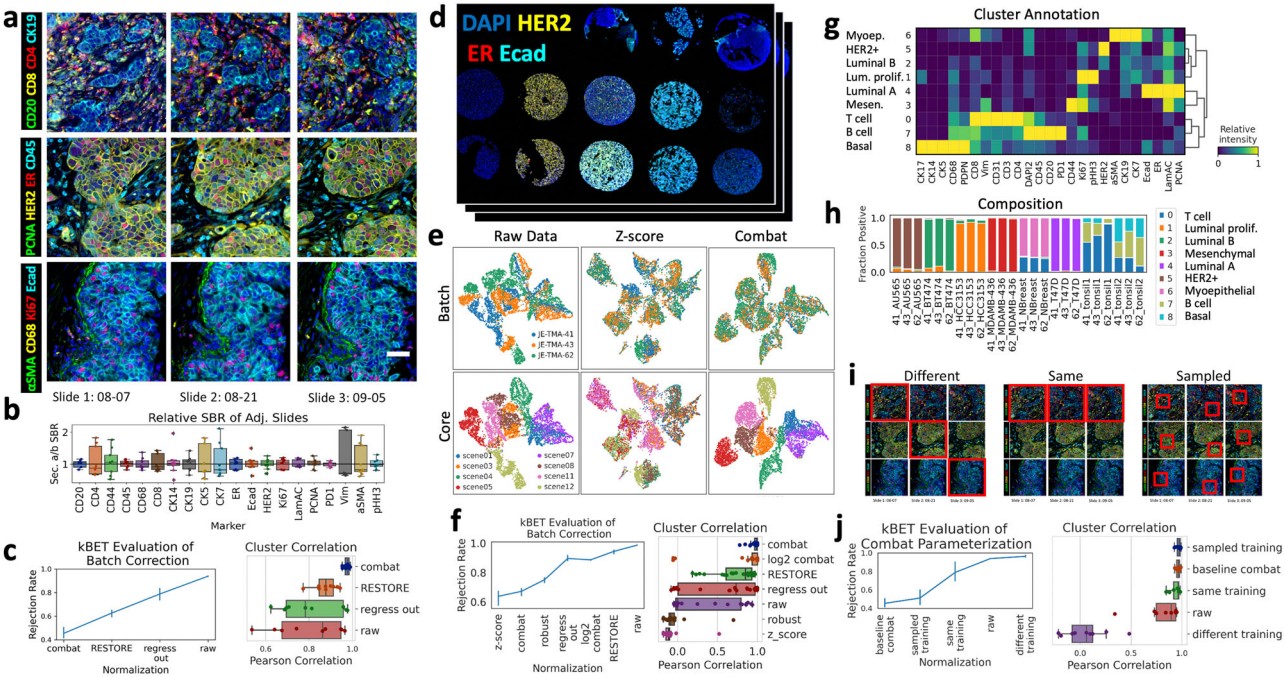

**Fig. 5 Reproducibility and normalization of CyCIF staining intensity. a** Representative images, generated with our python code, of 12 markers from three adjacent sections of a breast cancer TMA stained with a 20-marker CyCIF panel, scale bar = 50 μm. **b** Relative SBR of slide 1-2, 2-3, and 1-3 for each marker. **c** Evaluation of batch correction with kBET (left, lower rejection rates indicating better batch correction, n = 3 batches, 5400 cells) and correlation (Pearson, n = 9) of Leiden cluster composition between replicate cores for different batch correction methods (right). **d** Overview of breast cancer cell line and normal tissue TMA created to represent breast cancer subtypes. **e** UMAP projection based on single-cell marker intensity; left to right: raw data, z-score normalization and ComBat normalization, colored by batch (top) and TMA core/cell line (bottom). **f** Evaluation of batch correction with kBET (left, n = 3 batches, 7200 cells) and cluster correlation (right, n = 24) on cell line TMA. **g** Heatmap of relative mean intensity of each marker in the ComBat normalized, Leiden-clustered cell line TMA data, with annotation on left. **h** Fraction of cells in each cluster from (**g**), showing a similar composition of technical replicates and reflecting known normal tissue and cell line cell types. **i** Schematic of different selection methods to parameterize the ComBat algorithm. **j** Evaluation of ComBat parameterization with kBET (left, n = 3 batches, 7200 cells) and cluster correlation (right, n = 9). **c**, **f**, and **j**. Error bars = standard error of the mean, n = 3 slides.

**Reproducibility and batch normalization**. Following panel optimization, we assessed the reproducibility of CyCIF by analyses of three serial TMA sections stained with the same 20-antibody panel on different dates (Fig. 5a). We used manual thresholding to quantify SBR, as in our standard versus CyCIF analysis (Supplementary Fig. 13a, b, NB5-1). We compared our manual method of SBR calculation to the use of intensity quantiles to estimate dynamic range[10], and found that manual thresholds were more reliable when markers were negative, (e.g., HER2 in normal breast, Supplementary Fig. 14, NB5-2). With both methods, we found that the ratio of SBR between replicates for all 20 antibodies was close to one (relative SBR: mean = 1.02, SEM = 0.23, n = 20, Fig. 5b and Supplementary Figs. 13c, 14, NB5-2). Similarly, the difference in percent positive cells between each replicate was small (relative percent positive: mean = 1.10, SEM = 0.38, Supplementary Fig. 13c, NB5-2). These data demonstrate reproducibility between batches, with the SBR of each antibody being stable, despite variations in raw intensity values (Supplementary Fig. 9c, NB5-2). We used the same dataset to examine inter-patient variability on each TMA. The relative SBR between patients was often far from one (Supplementary Fig. 13c, NB5-2), likely reflecting a combination of biological variability and pre-analytical factors that affect antibody performance in FFPE tissues[26].

Since we observed intensity variation between CyCIF replicates, batch normalization was required for downstream analytics such as cell classification by unsupervised clustering. To quantify the effectiveness of batch correction methods on our serial TMA

sections, we used the kBET algorithm[27], which, given a k-nearest neighbor graph, compares batch-label distribution in random subsets of neighboring cells to the global batch-label distribution, thus quantifying how well-mixed batches are in technical replicates (Fig. 5a, c, and Supplementary Fig. 15, NB5-3). Previously, we developed RESTORE, which utilizes mutually exclusive marker expression to predict a background threshold[28]. For normalization, we set all values below the RESTORE threshold to zero and scaled those above the threshold between 0 and 1. We compared RESTORE normalization to ComBat[29] and mean-only correction (regress out), both implemented in scanpy[30]. ComBat produced the best batch correction (kBET = 0.45, SEM = 0.05, n = 3 slides, lower KBET rejection rates indicate better batch correction, with 0 being perfect mixing and 1 being no mixing of batches) with RESTORE also correcting some of the batch effect (kBET = 0.62, SEM = 0.03, Supplementary Fig. 15, NB5-3). Unsupervised Leiden clustering showed that both ComBat and RESTORE recovered known biology (Supplementary Fig. 15). To quantify the stability of clustering across batches, we calculated the Pearson correlation of the positive fraction of each cluster between each section (n = 9; 3 cores × 3 sections). ComBat normalization produced higher correlation with lower variance (indicating fewer cores with low correlation) than RESTORE normalization, but both were improvements over the raw data (ComBat mean Pearson correlation = 0.97, SEM = 0.01, RESTORE mean = 0.88, SEM = 0.05, raw mean = 0.8, SEM = 0.15, n = 9, Fig. 5c and Supplementary Fig. 15, NB5-4).

To further evaluate batch normalization methods, we created a TMA using FFPE breast cancer cell lines and normal tissues (Fig. 5d). Importantly, the cell lines were of known clinical and intrinsic subtype[31], establishing a ground truth for clustering results (Supplementary Table 3). Three TMA sections were stained using the same antibody-conjugates but in different panel orders (Supplementary Table 4). Images were processed through registration, autofluorescence subtraction, single-cell segmentation, and feature extraction. Due to different panel orders, raw data showed significant batch effects, with little mixing of cells from different TMAs in a UMAP projection based on single-cell marker intensity (Fig. 3e, NB5-5). Again, the kBET algorithm[27] and replicates' cluster correlation were used to quantify the effectiveness of batch correction methods (Fig. 5f and Supplementary Fig. 16, NB5-6). Thirty methods were evaluated on three random samples of 600 cells from each core in each replicate (Supplementary Fig. 16, NB5-5). Although Z-score normalization produced a reasonable kBET score (mean rejection rate = 0.64, SEM = 0.05, $n = 3$), the UMAP did not show separation of each core, i.e., known breast cancer subtypes (Fig. 5e, NB5-5). In contrast, the ComBat algorithm scored well with kBET (mean rejection rate = 0.67, SEM = 0.03, $n = 3$), separated the cores in the UMAP, and clustering recovered known cell types within cell lines and normal tissues (Fig. 5e, g, h). ComBat normalization applied without log2 transformation produced the most consistent clustering between replicate cores (mean Pearson correlation = 0.97, SEM = 0.06, 8 cores × 3 sections, $n = 24$; Fig. 5f, h and Supplementary Fig. 16, NB5-5). RESTORE normalization resulted in improved cluster correlation compared to raw data (mean = 0.73, SEM = 0.25 versus mean = 0.30, SEM = 0.41, $n = 24$), and recovered known biology (Fig. 5f and Supplementary Fig. 17, NB5-7); however, due to the lack of mutually exclusive marker pairs in cell lines, RESTORE did not perform as well as ComBat in this dataset (Fig. 3f).

Since the ComBat normalization method[29] scored highest in kBET batch correction and cluster correlation in multiple datasets while recovering biological features of interest, we tested its sensitivity to different input distributions for parameterization. Since ComBat standardizes the mean and variance across batches, the same control tissue must be present in each batch (Fig. 5i, j and Supplementary Fig. 16e, f). Sampling cells from three control tissues (i.e., sampled training) produced better kBET scores than using a single control tissue (sampled training with kBET = 0.51, SEM = 0.06 versus same training with kBET = 0.79, SEM = 0.07, $n = 3$, Fig. 5i, j and Supplementary Fig. 16, NB5-3). Cluster correlation varied with the resolution of clustering, indicating higher precision batch correction with sampling from three rather than a single tissue (with 9 clusters, sampled training mean = 0.98, SEM = 0.02 versus same training mean = 0.98, SEM = 0.04; with 17 clusters, sampled training mean = 0.97, SEM = 0.02 versus same training mean = 0.94, SEM = 0.05, $n = 9$ Fig. 5i, j and Supplementary Fig. 16f). Therefore, one or more control tissues with similar cellular composition to the sample tissues should be included in each batch of CyCIF for proper ComBat normalization. In the absence of control tissues and given mutually exclusive marker pairs, the RESTORE algorithm is a good alternative that produced a high correlation in cluster composition in multiple replicate CyCIF experiments (mean correlation = 0.88; 0.73). Additionally, in the absence of blank images for autofluorescence (AF) subtraction, RESTORE reduced the influence of AF on analytics (Supplementary Fig. 15).

In summary, we performed extensive validation of the CyCIF multiplex imaging method to optimize fluorophore signal removal, antibody order and autofluorescence subtraction. We also quantified CyCIF similarity to a standard IF and the reproducibility of staining, using this data to evaluate methods for batch normalization and cluster analysis to define biologically-relevant cell types. In parallel, we developed the open-source python library, mplexable, for reproducible image processing and analysis. We provided all of our code and data not only to document and reproduce our work, but to enable the use and further community development of our analytics (https://github.com/engjen/cycIF_Validation). Our validation studies and computational tools will facilitate the maturation of multiplex imaging methods toward quantitative, reproducible characterization of protein expression in intact tissue sections.

## Methods

**Tissue and cell button preparation.** We purchased TMA tissue sections (BR1506, US Biomax, Derwood, MD). All human tissue was collected under HIPPA-approved protocols with the highest ethical standards with the donor being informed completely and with their consent (OHSU Biolibrary IRB 4918). Tissue blocks and tissue microarrays were from archival tissues fixed with standard methods (Biomax, Derwood, MD and OHSU Biolibrary, Portland, OR). Cell lines were cultured in conditions following Neve et al.[31]. Cell line sources and culture media are as follows: AU565, ATCC, RPMI1640 + 10%FBS, BT474, ATCC, PMI1640 + 10%FBS, HCC1143, Adi Gazdar (now available through ATCC), RPMI1640 + 10%FBS, HCC3153, Adi Gazdar, RPMI1640 + 10%FBS, MDAMB436, ATCC, DMEM + 10%FBS, T47D, ATCC, RPMI1640 + 10%FBS. FFPE cell buttons were prepared as previously described[32]. For each cell button, cells were scraped from two 15 cm plates and spun down (in 10 ml of culture media, no serum, for 4 min × 1000 rpm) in collodion-coated 15 ml glass centrifuge tubes (Fisher #C408-500). Collodion bags containing pellets were gently removed from the tube, tied with thread, and transferred to 10% buffered formalin on ice. Cell pellets in collodion bags were fixed overnight at 4 °C in 10% buffered formalin, then transferred through graded ethanol series (30, 50, and 70%) for one hour each, and processed and embedded in paraffin. For cell line TMA creation, 1 mm cores from normal breast and tonsil (OHSU Biolibrary), or FFPE cell buttons were punched and inserted into the recipient block with the Estigen manual tissue arrayer MTA-1.

**Cyclic immunostaining.** Formalin-fixed paraffin-embedded (FFPE) human tissues were sectioned at 4–5 microns and mounted on positively charged slides (Tanner Adhesive Slides, Mercedes Medical, TNR WHT45AD). The slides were baked overnight at 55 °C (Robbin Scientific, Model 1000) and for an additional 30 min at 65 °C, (Clinical Scientific Equipment NO. 100). Tissues were deparaffinized and hydrated through xylenes and graded ethanol (EtOH) as follows: xylenes (3 × 5 min), 100% EtOH (2 × 5 min), 95% EtOH (2 × 2 min), 70% EtOH (2 × 2 min), and distilled and deionized water (ddH$_2$O, 2 × 5 min). Two-step antigen retrieval was performed in a Decloaking Chamber (Biocare Medical, Pacheco, CA) using the following settings: setpoint 1 (SP1), 125 °C, 30 s; SP2: 90 °C, 30 s; SP limit: 10 °C variation. Briefly, slides were placed in decloaking chamber in a plastic Coplin jar containing citrate buffer, pH 6 (10 mM citrate, Sigma C-1909). Two additional polyethylene Coplin jars with buffer were placed in the chamber to heat, which contained ddH$_2$O and 1x Target Retrieval Solution, pH 9 (Agilent S2367). The chamber was heated to 125 °C, held for 30 s (SP1), then cooled to 90 °C, 0 psi and held for 30 s (SP2). After the SP2 program was completed, the decloaking chamber was turned off, opened, and slides were dipped in the Coplin jar containing hot ddH$_2$O for ~1 s. Slides were then transferred to hot 1x Target Retrieval Solution pH 9. The lid was placed back on the chamber, and slides remained in hot pH 9 buffer for 15 min. Following this two-step antigen retrieval, the tissues were washed in two brief changes of ddH$_2$O (~2 s) and then washed once for 5 min in 1x phosphate-buffered saline (PBS), pH 7.4 (Fisher, BP39920).

**Pre-quenching to reduce autofluorescence.** Next, pre-quenching was performed on tissues to reduce tissue autofluorescence. Quenching solution containing 20 mM sodium hydroxide (NaOH) and 3% hydrogen peroxide (H$_2$O$_2$) in 1x PBS was freshly prepared from stock solutions of 5 M NaOH and 30% H$_2$O$_2$, and each slide was placed in 10 ml quenching solution. Slides were quenched face down on ~1 mm risers in a four-well rectangular tissue culture dish (each well holds one slide), under incandescent light, for 30 min. Lamps with 60-Watt incandescent bulbs were positioned so the bulb was four inches above the four-well dish. Placing slides in the outer two wells and leaving the center wells empty resulted in the temperature increasing from 23 °C to 39 °C over 30 min (see Supplementary Fig. 5c). Slides were then removed from the chamber with forceps and washed 3 × 2 min in 1x PBS. Sections were blocked in 10% normal goat serum (NGS, Vector S-1000) and 1% bovine serum albumin (BSA, Sigma A7906) in 1x PBS for 30 min at room temperature in a humidified chamber. Plastic coverslips (IHC World, IW-2601) were used to spread the blocking solution evenly across tissue. Tissues were soaked briefly in PBS in a Coplin jar to remove the plastic coverslip, then washed 1 × 5 min in PBS. Coverslips (Corning; 2980-243 and 2980-245) were mounted in Slowfade Gold plus DAPI mounting media (Life Technologies, S36938). Pre-staining autofluorescence signal was acquired using a Zeiss Axioscan

Z.1 (see imaging protocol below). After acquiring the autofluorescence signal, slides were soaked in 1x PBS for 10–30 min in a glass Coplin jar, waiting until the glass coverslip slid off without agitation.

**Primary antibody staining**. Primary antibodies were diluted in 5% NGS and 1% BSA in 1x PBS (see Supplementary Data 1) and applied overnight at 4 °C in a humidified chamber, covered with plastic coverslips (IHC World, IW-2601). Following overnight incubation, tissues were washed 3 × 10 min in 1x PBS and coverslipped as described above. Antibody information is provided in Supplementary Data 1.

**Fluorescence microscopy**. For standard versus CyCIF and reproducibility experiments, fluorescently stained slides were scanned on the Zeiss AxioScan.Z1 (Zeiss, Germany) with a Lumencor SpectraX-IR light source (Lumencor Inc., Beaverton, OR). The filter cubes used for image collection were DAPI (Semrock, LED-DAPI-A-000), AF488 (Zeiss 38 HE), AF555 (Zeiss 43 HE), AF647 (Zeiss 50), and Alexa Fluor 750 (AF750, Chroma 49007 ET Cy7). The exposure time was determined individually for each slide and stain, and the LED light intensity was fixed at 100%. Full tissue scans were taken with the 20x objective and stitching was performed in Zen Blue image acquisition software (Zeiss). After those initial experiments, a new light source was purchased and used for collecting quenching and round order data. For those experiments, a Colibri 7 light source (Zeiss), with the same filter cubes, except DAPI (Zeiss 96 HE), was used. The exposure time was determined individually for each slide and stain to ensure maximum dynamic range without saturation, and the LED light intensity was fixed at 10% (DAPI), 20% (AF488), and 50% (AF555, AF647, AF750). Full tissue scans were taken with the 20x objective (Plan-Apochromat 0.8NA WD = 0.55, Zeiss) and stitching was performed.

**Quenching of fluorescence signal for cyclic immunostaining**. After successful scanning, slides were soaked in 1x PBS for 10–30 min in a glass Coplin jar, waiting until the glass coverslip slid off without agitation. Quenching was performed as described above, in the section on Pre-quenching to reduce autofluorescence. After removal from the quenching solution, slides were washed 1 × 5 min in 1x PBS and subsequent rounds of primary antibodies were applied, diluted in blocking buffer as described in the section of Primary Antibody Staining.

**Image registration, autofluorescence removal, and segmentation**. Scanned images were first split into separate scenes using the function Split Scenes (Write files) in Zeiss Zen Blue software (with "Include scene information in the Generated File name" unchecked). For the datasets used in this work, we did not apply flat field correction, although it may be applied in Zen using the function Shading Correction. Using the same software, each scene was then exported to 16-bit grayscale uncompressed TIFF using the function Image Export. Quality control and metadata extraction were performed in python. TIFF images from each round of cyclic immunostaining were registered based on DAPI staining as follows. Image features were found with the detectSURFFeatures function, and automated feature matching was performed with the matchFeatures function, in Matlab (R2017B 9.3.0.713579, MathWorks, Natick, MA). Image registration was the performed using Matlab' estimateGeometricTransform function and performing affine registration (scaling, rotation, translation). Although Matlab was used for the majority of registration in this work, we also successfully registered images using a python implementation. Both scripts are provided (Matlab: https://gitlab.com/engje/mplexable/-/blob/master/mplexable/src/template_registration_mscene.m and python: https://gitlab.com/engje/mplexable/-/blob/master/mplexable/src/template_registration_mscene.py).

Autofluorescence subtraction preceded segmentation. Images of unstained tissue were acquired in each channel, before and after staining. For each marker, background images were scaled linearly by exposure time and relative round and subtracted using mplexable.

Deep learning based cell segmentation was performed with Cellpose, a generalist algorithm for cellular segmentation[33]. Cellpose was used to generate nuclear and cell masks by classifying pixels on the basis of a DAPI or E-cadherin antibody staining, respectively. The following parameters were used for Cellpose segmentation: for the cells, diameter = 30 pixels, flow_threshold = 0.6, min_size = 113; for the nuclei, diameter = 30, flow_threshold = 0, min_size = 28. Nuclei with no E-cadherin (Ecad) staining (i.e., non-epithelial cells) were expanded by five pixels (1.6 μm) to approximate the cytoplasm, based on the average measurement of immune cell cytoplasm in images. The cytoplasm was derived by subtracting the nuclei area from the cell segmentation result, or from the five-pixel expansion result in the case of Ecad negative cells. The mean intensity of each subcellular region was extracted using mplexable. Watershed algorithm-based cell segmentation was performed for some datasets, using in-house java software for the following operations. A z-projection of DAPI images from all rounds of staining was processed with the white top-hat algorithm that separates individual nuclear candidates from the background. Contours of nuclei were detected with the Prewitt operator, and single nuclei were segmented by applying a watershed algorithm to the nuclear contours, with top-hat candidates as seeds. Nuclear segmentation accuracy was improved by sorting nuclei based on the expression of

tumor cytokeratins or immune markers and using this information to set a maximum nuclear size for the watershed algorithm. If a cell was positive for cytokeratins, it was allowed to have a larger nucleus than cells that were negative for cytokeratins because it was assumed to be an epithelial cell. Cell segmentation was achieved by applying a watershed on the Ecad image with segmented nuclei as seeds, or by inflating the nuclei if the Ecad marker was negative. The resulting segmentation mask defined nuclear and cytoplasmic regions for each cell. Mean intensity used for downstream analysis was selected for each marker based on its biologically-relevant subcellular region (e.g., cytoplasm for CK19, nuclei for Ki67).

**Data analysis**. For single-cell analyses, single-cell mean intensity was used for clustering, as in Figs. 4 and 5. For percent positive calculation, as in Fig. 2c (left), cells with a mean intensity above threshold were considered positive. Tissue retention was calculated in Fig. 2e–h by thresholding DAPI using the Li algorithm[34] and considering cells above the DAPI threshold as retained in that round. For signal-to-background (SBR) calculations, mean intensity was integrated across the entire slide or region of interest, as in Fig. 2c, right, Figs. 3, and 5b. In fluorescence imaging, background adds to the signal of interest[35], so SBR was calculated as (mean intensity of positive pixels – mean intensity of negative pixels)/ mean intensity of negative pixels (NB2-1). For SBR quantification in Fig. 2, thresholds were applied directly to image data (no single-cell segmentation) and signal was taken as the mean pixel intensity above the threshold, while background was defined as the mean pixel intensity of pixels below the threshold and 30 pixels away (10 μm) from the positive pixels to exclude the influence of lateral bleed through. Since the threshold directly determines the result, thresholds were used that selected a similar pixel pattern and area in adjacent sections. The same marker in adjacent sections was visualized side-by-side, and the respective thresholds were adjusted until the positive pixels were as equivalent as possible, which was estimated by eye. Therefore, although the threshold reflected the subjective decision of the researcher, it allowed the comparison of similar pixels in replicates across adjacent sections. The masks that resulted from thresholding are provided for visual assessment of thresholds (https://github.com/engjen/cycIF_Validation/blob/master/Extended_single_vs_cyclic.ipynb and https://github.com/engjen/cycIF_Validation/blob/master/Extended_Reproducibility_3TMA_Tissue.ipynb).

We tested whether the same threshold could be applied to different regions of the same slide by measuring the correlation between two ROIs in normal breast and two ROIs in HER2 + breast tumor given the same threshold. The mean fluorescence intensity measured above a threshold and the intensity of background noise were highly correlated between ROIs (Supplementary Fig. 2). Finally, we tested whether manual thresholds gave us a different answer for SBR calculations than estimating SBR at the 95th/5th quantile (Supplementary Fig. 14).

Dynamic range was estimated using the 4th and 99.5th quantile of mean intensity for markers in tissues with known positive staining. We compared different ranges for estimating dynamic range (5th to 98th percentile versus 5th to 99.9 percentile, see Supplementary Fig. 14). We found that using a higher maximum did not change the dynamic range as much for common markers (e.g., CK7) but had more effect on rare markers (e.g., Ki67 and alpha-SMA). Therefore, we selected the 99.5th percentile as the maximum to reflect both common and rare markers' dynamic range. In cases where we did not set a manual threshold, SBR was estimated as the ratio of the (99.5th quantile – 4th quantile)/4th quantile of mean intensity.

For SBR quantification in Fig. 3, we needed to calculate SBR in the presence of artifacts such as nonspecific staining, bleed through, autofluorescence and incomplete quenching. We utilized the napari image viewer to overlay all of the markers, set thresholds, and create masks. For the positive signal, we set a threshold (recorded in our 20211007_napari.py script) that created a mask including all of the positive staining plus any bright artifacts. We then manually erased areas of the mask covering imaging or staining artifacts and saved the mask for future reproducibility. For the background signal, we manually selected six regions of the image exhibiting background caused by the artifacts listed above and again saved the mask. We then extracted the mean intensity of the positive and background areas of the image and used these to calculate SBR as described above.

For F1 score calculation in Supplementary Fig. 11, we again used the napari image viewer to overlay staining, segmentation results, and positive cells based on manual thresholding. Based on the staining pattern and other marker's expressions (e.g., membranous CD8 staining in CD45 + cells was a true CD8 positive), we manually annotated false negatives in three 2000 × 2000-pixel ROIs. False positives were any cell with AF488 autofluorescence >1024-pixel intensity, true positives were cells above this threshold excluding false positives and true negatives were all other cells neither positive, false positive or false negative.

Normalization methods tested included transformations (raw, log2, or arcsinh), division by background signal, determined either with RESTORE, which requires mutually exclusive marker expression in different cell populations, or the third quantile of background fluorescence measured in the reverse subcellular compartment of expected localization (e.g., cytoplasmic signal for nuclear markers), scaling methods (standard, min-max, max-abs, robust, quantile, and power) which are implemented in the python library scikit-learn[36] and batch correction algorithms regress_out and ComBat, implemented in the python library scanpy[30]. For RESTORE normalization, the scikit-learn[36] TruncatedSVD decomposition function was used to quantify the L-shaped distribution of marker

pairs; in each core, markers were considered mutually exclusive if the function yielded an R-value above 0.5, or above 0.2 when no pairs reached the 0.5 cut-off. For global thresholds, a more stringent cutoff was used (for the presented TMA datasets 0.66 globally or 0.5 when no pairs reached the 0.66 cut-off. For each core or batch, this procedure generated data-driven mutually exclusive marker pairs. The selected mutually exclusive marker pairs were used to calculate RESTORE thresholds, and the median threshold produced by multiple marker pairs was selected for normalization. Mean intensity was normalized with division by local (per core) and global (per batch) thresholds. For the RESTORE scale method, cells with intensity below the threshold were set to a random value between 0 and 0.02, while all cells with intensity above threshold were scaled to a range of 0.02–1 for each marker.

Following normalization, 7200 cells (cell lines) or 5400 cells (tissues) were randomly sampled (i.e., 600 cells per TMA core from each batch) and evaluated for batch effects using the kBET algorithm[27] ($n = 3$). UMAP visualization and graph-based Leiden clustering (resolution = 0.6) was carried out using scanpy[30].

**Statistics and reproducibility**. All visualizations and results can be fully reproduced from the raw images with the accompanying code and data, here https://github.com/engjen/cycIF_Validation. Statistical analyses were conducted in python using the scipy[37] library. Replicates were defined as separate CyCIF experiments. Sample sizes are defined by the number of tissues stained, including tissue cores in TMAs.

**Reporting summary**. Further information on research design is available in the Nature Research Reporting Summary linked to this article.

## Data availability

All image data is available at synapse.org, at cycIF_Validation, syn23644107. The synapse platform requires registration. With a free account, the images are freely accessible. All other source data used to produce the graphs and figures is available here: https://github.com/engjen/cycIF_Validation, https://doi.org/10.5281/zenodo.6049278.

## Code availability

The data, code and Jupyter notebooks to reproduce the analyses herein are at https://github.com/engjen/cycIF_Validation, https://doi.org/10.5281/zenodo.6049278. For image processing, we developed mplexable, available through the Python Package Index, https://pypi.org/project/mplexable/. An image processing tutorial with Zeiss Axioscan example images is available at https://www.synapse.org/#!Synapse:syn26958265. Additionally, we demonstrate processing Zeiss Axioscan, Akoya CODEX, and Miltenyi MACSima prototype images in our pipeline example Jupyter notebooks, here: https://gitlab.com/engje/mplexable/-/tree/master/jupyter.

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

## Acknowledgements

We would like to thank former lab members Dr. Moqing Liu, Aesoon Bensen, and Lydia Grace Campbell as well as former interns Clayton Peltz and Cole Meyer for assistance with staining and imaging. We would also like to thank Damir Sudar for technical support related to image storage. We appreciate the sample scanning assistance of both Drs. Stefanie Kaech Petrie and Crystal Chaw at the OHSU Advanced Light Microscopy Core. The OHSU Biolibrary and OHSU Histopathology Share Resource provided tissue blocks and histology services. We would like to thank Elliot Gray for assistance with python and shell scripts. We would like to thank Drs. Andreas Bosio and Werner Muller for insightful discussions on the Miltenyi MACSima prototype image analysis. The Miltenyi MACSima Prototype instrument was provided by Miltenyi. Drs. Young Hwan Change and Erik Burlingame contributed ideas for data analysis and image visualization as well as code for RESTORE normalization and registration. Dr. Guillaume Thibault provided the code for watershed segmentation. This work was supported by funding from the Prospect Creek Foundation, Susan G. Komen Foundation, and OHSU Foundation. Additionally, J.W.G acknowledges support from NIH/NCI U54 CA209988, NCI SBIR 1R44CA224994-01, and NIH/NCI U2C CA233280. The OHSU Knight Cancer Institute Shared Resources are supported by the Knight Cancer Institute Cancer Center Support Grant (NIH/NCI P30CA69533).

## Author contributions

J.E., S.L.G., K.C., and J.W.G. conceived of the project. J.E and K.C. designed the antibody panel, performed experiments and processed image data. J.E. and E.B. developed free and open-source python tools and Jupyter notebooks for image processing, visualization, and analysis. Z.H. performed staining and imaging experiments. T.Z. established quenching and antibody labeling protocols. J.E. drafted the manuscript and prepared the figures. S.L.G., K.C., and J.W.G. directed the project and edited the manuscript.

## Competing interests

J.W.G. has licensed technologies to Abbott Diagnostics; has ownership positions in Convergent Genomics, Health Technology Innovations, Zorro Bio, and PDX Pharmaceuticals; serves as a paid consultant to New Leaf Ventures; has received research support from Thermo Fisher Scientific (formerly FEI), Zeiss, Miltenyi Biotech, Cepheid (Danaher), Quantitative Imaging, Health Technology Innovations, and Micron Technologies; and owns stock in Abbott Diagnostics, AbbVie, Alphabet, Amazon, Amgen, Apple, General Electric, Gilead, Intel, Microsoft, Nvidia, and Zimmer Biomet. S.L.G. has received research support from Quantitative Imaging. The remaining authors declare no competing interests.
