## [Peer Review File · Communications Biology]

Reviewers' comments:

Reviewer #1 (Remarks to the Author):

The manuscript entitled "A framework for multiplex imaging optimization and reproducible analysis" is describing both experimental optimization and software pipeline for multiplexed CyCIF imaging. Authors provided alternative CyCIF protocols to improve fluorophore quenching, to remove autofluorescence and to design antibody panels. In addition, Eng et. al. also developed serious python scripts and library to handle image processing, data normalization and reproducibility test.

As an emerging field of tissue multiplex imaging with various methods developed, many issues and challenges remain. Authors ambitiously addressed a few fundamental issues: reproducibility, batch normalization and signal-to-background of antibodies. However, most findings and conclusions in this manuscript are either lack of enough data to support with, or previously reported. For example, while authors argued their pipeline could improve reproducibility, only a few (n=3) samples were tested with limited analyses, as lack of sufficient statistical supports. The protocol optimization on dye quenching, while interesting, the results were not convincing while missing direct comparison and quantitative analysis. I did appreciate the thoroughness of this study to address several important aspects in tissue imaging methods, but it would be much improved if they could focus on one area (either experimental or computational) with deeper insights. My specific comments and questions are listed below.

Major comments:

1. The whole workflow and codes were customized for specific image modality, and it might require certain amount of rewriting and tweaking in order to be used by others. It would be better to consider at least to repackage these individual modules with containers, pipeline language or GUI. Thus, more researchers in the field could find these tools useful.
2. The AF subtraction method described here is not totally novel, has been mentioned by several other multiplexed platform (eg. CODEX and Ultivue). Also, while linear decay of AF observed in this sample, It might not be generalizable. As there are many other sources of AF, like RBC and pigments, these could behave differently upon quenching. Furthermore, the AF subtraction only showed marginal difference, as shown in Ext Fig 5, with few markers. Why? Any possibility of overcompensation and over-subtraction?
3. Image registration described in Methods was done with MATLAB, but in the mplex-image codes, there is a python registration module. Which one was used? If indeed MATLAB was used, how it has been incorporated in the pipeline? Did you provide the MATLAB codes for registration?
4. As the manual gating was used for defining thresholds in pixel level, how could you ensure the reproducibility among various researchers and sites? Could you provide an example of gating procedures, as well as the guidelines you used?
5. The panel optimization in cyclic staining protocol is important. The authors provide only one example here. I still do not quite understand the principle behind. Could you elaborate more on the method you used?
6. The issue with accessing sample images (I couldn't access through Synapse). Without that, readers (including myself) could not evaluate the pipeline and tools provided.

Minor comments:

1. Page 3 line 83: "Increase H2O2 concentration above 3% did not improve quenching rate....." any quantitative results to support? Also, do you increase the NaOH concentration with 4.5% & 6% H2O2.
2. Page 3 line 85: "Gentle heating with....": not clear how to archive this, no description in the method section.
3. Page 6 line 145: "Bleed-through was evident for weakly staining and strongly staining antibodies..." The example you provided (PD1-AF647 & CK19-AF750) is quite intriguing. Because the AF750 and AF647 should be spectrally separately while using right filter sets. There might be a possibility that the chemical and physical factors (eg. conjugation or oxidation) could change spectrum properties of these photophores. It might need to be tested carefully.
4. Page 4 line 94: "...reduce overall tissue autofluorescence by roughly 25%.". Data/figure to support this point?

5. Ext. Data Figure 1. Vimentin & CD44 shows lower mean intensity compared to standard IF, relative to CycIF? Any explanation?
6. Missing catalog numbers & clones in Supplementary table 1
7. In Figure 2j Looks like different regions for left and right panels, right? If so, this might not be a fair comparison since different regions could have different background/AF. It would be better to find the same ROIs, using adjacent sections. Also, in ExtFig 6, the CD45 in R2 & R11 show similar nuclear backgrounds. Why it's different?

Reviewer #2 (Remarks to the Author):

This paper reports an improved approach for tissue image analysis that is particularly applicable for highly multiplexed immunofluorescence (IF) imaging of the spatial distribution of multiple protein biomarkers. Although the paper is focused on the analysis of multiplex IF performed using the multiplex imaging platform, CyCIF, the method should be generalizable to other multiplex methods.

Overall, this is an important contribution to the tissue image analysis field and will provide significant impetus for using methods like CyCIF to perform more robust image analysis of highly multiplexed tissues. The work establishes new protocols for removing autofluorescence background signals, order of antibody application for CyCIF, along with the development of several powerful, open-source algorithms for image processing and analysis.

The paper is suitable for publication after a few minor points are addressed:

Line 51-53. "Although we focus on validation of a single platform, our tools for image analysis facilitate quantitative and reproducible characterization of tissues by any multiplex imaging platform."

The authors should reference here a few other platforms besides CyCIF where their approach to image analysis of highly multiplexed proteins might be applicable and give further discussion how their methods can be used for these cases. Example include the recent use of photocleavable mass-tags in conjunction with MALDI-MSI (Yagnik, G. et al. J Am Soc Mass Spectrom 32, 977-988); Imaging mass cytometry (Giesen, C. et al. . Nat. Methods 2014, 11, 417-422) and in the field of transcriptomic combinatorial labeling where it is now possible to image the location of thousands of RNA species (Chen, K.H. Science 215 348 (6233, aaa66090).

Line 64: We benchmarked our CyCIF protocol against a direct IF protocol. CyCIF staining in normal and malignant breast tissue was compared to standard direct IF on adjacent tissue sections.

Since this is a major goal of the study, a summary is needed later in paper which not only compares the analytical results of the two approaches but a comparison of the two different work flows. For example, if equivalent results can be obtained using adjacent tissue sections, what advantages are there for performing CyCIF on the same tissue section (e.g. shorter overall processing time, etc.)?

Line 93: "to reduce overall tissue autofluorescence by roughly 25%."

Is this reduction in autofluorescence true for all of the tissue types analyzed? Can the authors give more specific information (e.g. did it work for most of the different tissues studied?)

Line 118. We implement both the "scaled" and "baseline" algorithms for autofluorescence subtraction in mplex-image.

Can the authors provide more guidance on when to implement scaled vs. baseline algorithm for autofluorescence subtraction?

Line 129-132: Non-specific nuclear staining was observed for several antibodies conjugated to the AF750 fluorophore, and unexpectedly, it diminished on the second application (Fig. 2j,

Can the authors provide some suggestions why this might occurred?

Figures: Define in legend the scale (e.g. scale bar needed) for the tissue images shown throughout (e.g. Figure 1f, 2a, 2e, 3b)

Reviewer #3 (Remarks to the Author):

Eng et al. in this paper address an actual problem, the need to evaluate through precise and shared image analysis methods the performances of multiplexing methods in order to optimize the method itself and produce results that are the real reflection of the biological processes investigated. With this work they share as an open source their python code (mplex-image) to offer a fully-reproducible image analysis pipeline that should allow the user to quantitatively assess antibody labeling and evaluate strategies for signal removal, antibody specificity assessment, background correction and batch normalization. They develop and apply this method using the CyCIF multiplex imaging method, offering optimization and validation to the method itself. It is true that their work has the potential to raise awareness to the need of validation studies for multiplexing methods (that too often claim performances that raise expectations that are not met in the laboratory practice). Also, offering the tools as an open source it is certainly a big plus. Nevertheless, this paper doesn't meet its full potential because there are several points that could be improved.

A first very general comment is that the authors present a missing need in the literature related to multiplexed IF analysis. Most of the analyses are performed in only one type of tissue/cancer, and we have concerns about how generalizable their results are. In our experience, multiplexed images tend to be very heterogeneous (tissue type, pathology, Abs used, etc.) so what is true for pancreatic tissue might not be true for breast tissue.

Looking specifically at the main text:

1) the authors use their pipelines to validate and optimize CyIF.

Line 37 and Line 52: Very limited literature about iterative fluorescence methods, a lot of them are not listed here. For completeness' sake the authors should research literature and list all the IF-based cyclic methods: do the authors foresee differences applying their framework to the other methods? How the pipeline may need to be adjusted?

2) the authors do benchmarking with standard IF (visual inspection + SBR + positive cells count) and they conclude that the SBR and specificity of antibody staining in the first five cycles of CyCIF is similar to standard IF. Moreover, they detect tissue loss and point to the need for gentler quenching, identifying quenching using 3% H₂O₂ for 30 minutes with incandescent light + an additional round of pre-quenching as the best modification for the method (though the pre-quenching is not novel, as they admit).

Line 66: how could the authors then validate markers that are negative in the section of interest? Do they expect to always have a positive control inside the reference tissue? Were these negative markers discharged by default?

Line 67: the authors talk about "quantification of the signal" using a threshold, but this is not quantification of the signal, it's a count of cells considered as "positive" without taking in consideration the full range of value of the signal. Moreover, thresholds were set manually in small ROIs. I have concerns about how generalizable these thresholds are to other regions of the images.

Line 68 (NB1-2,

https://github.com/engjen/cyCIF_Validation/blob/master/Extended1_single_vs_cyclic.ipynb): the authors show various images heavily affected by the field-of-view-artifact. However, they do not mention any flat-field-correction for their images throughout the MS. This can affect the results obtained by antibodies with low SNR.

Line 59 and 75: the authors state that 40-60 protein could be labelled (10-15 cycles) before tissue

deterioration, but also that SBR and antibody specificity is similar in CyCIF and standard IF for 5 cycles: how can these two sentences coexist? How valuable can be the comparison with standard IF be for markers that are stained in later cycles?

Line 78: the authors talk about "increased tissue loss", do they specifically mean physical detachment of the tissue from the slide? Loss of antigenicity of the tissue? Or both? From the images in Extended Data Fig. 1a we can assume both. The legends of the images are not sufficiently detailed: what are we supposed to see from the images that they provide? There is always written what the graph/image is about, but not a description of what it is meant to show.

3) Subsequently, the authors attempt an original approach on autofluorescence: they investigate single-cell autofluorescence (AF) changes and highlight that one of the AF components following linear changes - based on this the authors implemented a "scaled" next to a "baseline" algorithm for autofluorescence subtraction in mplex-image.

Line 102: "Overlaying the cells in each cluster on the tissue revealed cells in cluster 7 comprised tissue structures with bright autofluorescence, such as elastic fibers" → elastic fibers should not be nucleated structures. This evidence makes the segmentation approach used in this paper not believable since it includes measuring of the background rather than of the real cytoplasm of the cells. Moreover, figure 2e it is not clear in showing why the authors are sure that those structures are elastic fibers - the images are too small and not well readable.

Line 106-108: based on the results shown in figure 2.g, the assumption that AF decays linearly for other components seems to be oversimplifying. Image intensities are affected by a large series of variables including: noise caused by imperfections in image acquisition (see: <https://ieeexplore.ieee.org/document/5590292>) This might affect markers with low SNR where true positives are slightly above background.

Line 113: the authors give the definition of what they consider as false positive based on measuring AF at single cell level, but what do they consider as "false negative"? No definition is given.

4) A very nice paragraph in the paper is where the authors measure sensitivity and specificity of the antibodies in relation to the round in which they are performed (early vs late) in order to optimize the panel design following the results. This is definitely a fundamental concept for all cyclic methods.

Line 122: regarding Supplementary table 2, the rounds considered as "early" are until round 10 - yet, since at the beginning it was stated that the images are comparable to the gold standard until round 5, we wonder if it was a good strategy to include 10 rounds in the definition of "early" rather than limit the validation to less antibodies stained twice in a total of 10 rounds in total (5 early and 5 late).

5) Finally, the authors perform a reproducibility test: they stain 3 consecutive sections of a TMA on 3 different dates and compare two methods to evaluate the dynamic range of the stainings. With both they obtain similar results, highlighting reproducibility of the method but also the need for batch effect correction. They then compare three batch effect methods. All improved the clustering. Combat was found to be better, even though it needs the presence of control tissue, while RESTORE works better with mutually exclusive markers and does not need control tissue. This was done on tissue sections and repeated on cell line to check cell type identification.

Some more remarks regarding the methods:

Line 324: "Nuclear segmentation accuracy was improved by sorting nuclei based on expression of tumor cytokeratins or immune markers." This sentence is not clear – cytokeratins are cytoplasmic stainings, how can a cytoplasmic staining improve the segmentation of the nuclei? Please explain.

Line 326 – cell segmentation: how is the 5 pixel expansion justified? Why 5 pixels and not 2 or 10?

Line 327: "membrane" written here gives the false impression that the authors evaluate also the membrane location like a third subcellular area, while later it becomes clear they subsegment in nuclei and cytoplasm.

Line 342: "all cells with a mean intensity above threshold were considered positive": how were these thresholds calculated? How was the decision made?

Line 344: Not all the cells have the same background/foreground and therefore not the same SBR. Moreover, since positive/negative pixels are selected based on manual thresholds, they have control on the estimation of the SBR.

Line 348: How does the estimation of the dynamic range in the q5-q95 range affect rare markers expressed in very few cells (~0.1%)?

Dear Reviewers,

Thank you for the positive and constructive comments on our manuscript entitled “*A framework for multiplex imaging optimization and reproducible analysis.*” We have addressed all the questions and concerns, and are pleased to provide our revised and improved manuscript as well as the following point-by-point revisions to address the specific reviewers’ comments. These changes are also shown in red text in the revised manuscript to aid in re-review.

Reviewer #1:

We would like to thank the reviewer for the positive and constructive comments on our manuscript. We have made revisions to address the reviewer’s suggestions that are outlined below in a point-by-point manner, which we believe improve the clarity and impact of the manuscript.

General Reviewer #1 comment: *The manuscript entitled “A framework for multiplex imaging optimization and reproducible analysis” is describing both experimental optimization and a software pipeline for multiplexed CyCIF imaging. Authors provided alternative CyCIF protocols to improve fluorophore quenching, to remove autofluorescence and to design antibody panels. In addition, Eng et. al. also developed serious python scripts and library to handle image processing, data normalization and reproducibility testing.*

As an emerging field of tissue multiplex imaging with various methods developed, many issues and challenges remain. The authors ambitiously addressed a few fundamental issues: reproducibility, batch normalization and signal-to-background of antibodies. However, most findings and conclusions in this manuscript either lack enough data to support the conclusions or were previously reported. For example, while the authors argued their pipeline could improve reproducibility, only a few (n=3) samples were tested with limited analyses, thus this conclusion is lacking sufficient statistical support. While the protocol optimization on dye quenching is interesting, the results were not convincing and were missing direct comparison and quantitative analysis. I did appreciate the thoroughness of this study to address several important aspects in tissue imaging methods, but it would be much improved if they could focus on one area (either experimental or computational) with deeper insights. My specific comments and questions are listed below.

General Comments to Reviewer: We appreciate the reviewer’s point on our data size, which has now been increased in our resubmitted manuscript. Our reproducibility data, which is now in Figure 5, shows two separate experiments, each containing three tissue microarrays (TMAs). For kBET analysis of batch effects, we analyzed 20 markers in 5400 cells from 9 tissues and 24 markers in 7200 cells from 24 tissues (see Figure 5c, 5e, 5f, and 5j). When evaluating the reproducibility of clustering results across replicates, we considered the cluster composition in each tissue core, for n=9 and n=24 (Figure 5c, 5f, and 5j). For dye quenching, our quantitative analysis is now shown in Figure 3d and extended Figures 4 and 5.

Major Comments:

1. The whole workflow and codes were customized for a specific image modality, and it might require a certain amount of rewriting and tweaking to be used by others. It would be better to consider at least to repackaging these individual modules with containers, pipeline language or GUI. Thus, more researchers in the field could find these tools useful.

Comments to Reviewer: We agree and have developed our original mplex_image pipeline to be more customizable. The new version has been released as jinxif. For example, all lab- and platform-specific variables are configured in a single location (<https://gitlab.com/engje/jinxif/-/blob/master/jinxif/configure.py>) allowing for adaptation of the pipeline to different modalities. As an example of this, we provide Jupyter notebook templates for processing our Zeiss Axioscan images, Akoya CODEX images and Miltenyi MACSima images. Our main implementation of the

pipeline is in python for ease of customization and sharing, with utilization of Jupyter notebooks as a GUI-like interface.

Revisions Made: To highlight these changes, we have added the following text to the Code Availability section:

“The data, code and Jupyter notebooks to reproduce the analyses herein are at https://github.com/engjen/cycIF_Validation. For image processing, we developed jinxif, available through the Python Package Index, <https://pypi.org/project/jinxif/>. We demonstrate processing Zeiss Axioscan, Akoya CODEX and Miltenyi MACSima images in our pipeline example Jupyter notebooks, here: <https://gitlab.com/engje/jinxif/-/tree/master/jupyter>.”

2. The autofluorescence (AF) subtraction method described here is not totally novel as it has been mentioned by several other multiplexed platforms (e.g., CODEX and Ultivue). Also, while linear decay of AF was observed in this sample, it might not be generalizable. As there are many other sources of AF, like RBC and pigments, these could behave differently upon quenching. Furthermore, the AF subtraction only showed marginal difference, as shown in Ext Fig 5, with few markers. Why? Any possibility of overcompensation and over-subtraction?

Comments to Reviewer: To address the reviewer’s questions, we collected two additional datasets on AF during CyCIF, totaling 246 additional tissues and extending the experiment to a full 10 rounds of staining. As discussed below, although we saw different intensities of autofluorescence in different tissues, they followed the same trend upon quenching. In Extended Data Figure 11 (formerly Extended Data Figure 5) we compare two algorithms for AF subtraction. The “scaled” algorithm is marginally better in markers tending to have low SNR ratio (e.g., PCNA, CD8), but either algorithm improves over no AF subtraction. For example, in Figure 4h, you can see that the intensity of autofluorescence after pre-quenching is >2000 in some cells, which is greater than the expression of some markers in the panel (e.g., Extended Data Figure 6f). In order to avoid over subtraction, we recommend subtracting the autofluorescence images collected after round three or four, which have the lowest intensity. In this way, one can subtract between 60 and 100% of autofluorescence, but never more than 100%.

Revisions Made: Additional data was collected and analyzed, shown in the new Figure 4d-4j as well as Extended Figures 10, 11 and 12. The following text was also added to section Autofluorescence Dynamics and Corrections:

“To confirm these results, we collected two additional datasets, three adjacent sections from a 72-core TMA containing normal and tumor tissues (Fig. 4d) and an 11-core TMA containing HER2+ breast cancer tissues, quenched for a total of 10 rounds, 330 minutes and 6 rounds, 210 minutes, respectively. We segmented cells to obtain single-cell autofluorescence values, batch normalized and clustered as described for the pancreas tissue (Fig. 4e-4g, Extended Data Fig. 10). Although the AF488 autofluorescence initially declined in these datasets, as had been observed in the pancreas, it went back up, globally across all clusters (Fig. 4h, Extended Data Fig. 10). We did observe that the autofluorescence clusters correlated with one or more tissue types and pathologies (Fig. 4j, Extended Data Fig 11.), but they all showed the same trend, with a minimum intensity at three or four rounds of quenching (after pre-quenching, Fig. 4h, Extended Data Fig. 10). We saw similar trends in AF555 autofluorescence (Extended Data Fig. 10, 12). Therefore, to computationally remove autofluorescence without over-subtracting, we recommend collecting

“baseline” autofluorescence images after quenching at round 3 or 4, and subtracting, scaled by exposure time, from AF488, AF555 and AF647 channels. The AF750 channel does not have autofluorescence. This procedure will remove 60 – 70% of autofluorescence from the brightest rounds without subtracting any signal from the rounds with less autofluorescence (Fig. 4i, Extended Data Fig. 10, 11). Since different tissues and other multiplex imaging platforms may exhibit different autofluorescence dynamics, we implement both the “baseline” algorithm, which subtracts a single autofluorescence image, and a “scaled” algorithm, which assumes linear increase or decrease in autofluorescence, for autofluorescence subtraction in jinxif. We found that the scaled algorithm, by linearly interpolating autofluorescence between two background images, did reduce false positives for some markers in our panel (Extended Data Fig. 12). Since the brightest areas of autofluorescence showed a linear decrease across the first few rounds of CyCIF, the scaled algorithm is most advantageous in tissues with strong autofluorescence that are stained only a few rounds. Conversely, the “baseline” algorithm is simple, requiring collection of just one background image, and can remove the majority of autofluorescence background without over-subtraction in various tissue and experimental contexts.”

3. The image registration described in the Methods was done with MATLAB, but in the mplex-image codes, there is a python registration module. Which one was used? If indeed MATLAB was used, how is it incorporated in the pipeline? Did you provide the MATLAB codes for registration?
Comments to Reviewer: We used the MATLAB script because we have found it to be more successful in registering diverse tissues. In support of this, the MATLAB script is now provided in our new jinxif software. However, because some users will not have access to proprietary software, we also provide the python registration as a second option.

Revisions Made: We added the following text to the methods section:

“Although Matlab was used for the majority of registration in this work, we have also successfully registered images using a python implementation. Both scripts are provided for flexibility and ease of use (MATLAB: https://gitlab.com/engje/jinxif/-/blob/master/jinxif/src/template_registration_mscene.m and python: https://gitlab.com/engje/jinxif/-/blob/master/jinxif/src/template_registration_mscene.py).”

4. Since manual gating was used to define thresholds in pixel level, how can you ensure the reproducibility among various researchers and sites? Could you provide an example of gating procedures, as well as the guidelines you used?

Comments to Reviewer: In the revised manuscript, we ensure reproducibility of our work by providing the thresholds and masks necessary for readers to reproduce the findings reported herein.

Revisions Made: The following text added to the methods section to explain the role of manual gating in our studies:

“Since the threshold directly determines the result, thresholds were used that selected a similar pixel pattern and area in adjacent sections. The same marker in adjacent sections was visualized side-by-side, and the respective thresholds were adjusted until the positive pixels were as equivalent as possible, which was estimated by eye. Therefore, although the threshold reflected the subjective decision of the researcher, it allowed comparison of similar pixels in replicates across adjacent sections. The masks that resulted from thresholding are provided for visual assessment of thresholds

(https://github.com/engjen/cycIF_Validation/blob/master/Extended_single_vs_cyclic.ipynb and https://github.com/engjen/cycIF_Validation/blob/master/Extended_Reproducibility_3TMA_Tissue.ipynb).

5. The panel optimization in the cyclic staining protocol is important. The authors provide only one example here. I still do not quite understand the principle behind. Could you elaborate more on the method you used?

Comments to Reviewer: We have now expanded this data set which is presented in the new Figure 3 to give a more thorough explanation of our panel optimization. Additionally, in support of our conclusions we also show optimization using another platform, the Miltenyi MACSima.

Revisions Made: See our new analyses in Figure 3e-3i; and the new data in Extended Figure 8h-8k. The following text was added to the section Antibody Order Evaluation and Improvement:

“Thus, we designed an improved panel order which positioned antibody-conjugates tending to have non-specific staining in a later round (Fig. 3e), as well as addressed pitfalls of autofluorescence, channel bleed through and incomplete quenching (Fig. 3f-3h). We compared the original and optimized orders on near-adjacent slides cut from a TMA containing three tissues with positive staining for each marker. Signal-to-background ratio (SBR) quantification was completed by applying a threshold to each marker to find positive pixels, and manually selecting areas of non-specific background (e.g., tumor nests for immune markers and stromal areas for tumor markers). Our new panel order significantly increased SBR (Fig. 3i, NB2-6).”

6. There was an issue with accessing sample images (I couldn't access through Synapse). Without that, readers (including myself) could not evaluate the pipeline and tools provided.

Comments to Reviewer: The synapse platform requires registration (<https://www.synapse.org/#!RegisterAccount:0>). With a free account, the images are readily accessible.

Revisions Made: To ensure readers understand how to access the sample images, we added the following text to the Data Availability section:

“The synapse platform requires registration for access. With a free account, the images are readily accessible.”

Minor Comments:

1. Page 3 line 83: “Increase H₂O₂ concentration above 3% did not improve quenching rate....” any quantitative results to support? Also, do you increase the NaOH concentration with 4.5% and 6% H₂O₂.

Comments to Reviewer: We did not increase the NaOH concentration, as we did not want to change two variables simultaneously. For all quenching experiments, we quantified mean intensity of the stained versus the unstained tissues. Because of autofluorescence, we consider quenching complete when the intensity of the stained tissue reached baseline autofluorescence of the blank control.

Revisions Made: In the revised manuscript we added quantitative results shown in Figure 3d; Extended Figure 4 and 5. We also added a figure showing direct comparison of H₂O₂ percentage in Figure 3d and Extended Data Figure 4d. The following text was added to Quenching Condition Assessment section to explain the new data:

“Increasing H₂O₂ concentration from 3% to 4.5% or 6% in 20 mM sodium hydroxide did not improve quenching rate, and additional time in H₂O₂ also failed to completely eliminate strong signal (Fig. 3a, 3b, 3d Extended Data Fig. 4, 5 NB2-1).”

2. Page 3 line 85: “Gentle heating with....”: not clear how to achieve this, no description in the method section.

Comments to Reviewer: We added a more detailed explanation of the heating used during quenching to the methods section.

Revisions Made: The following text was added to the methods section:

“Slides were quenched face down on ~1 mm risers in a 4-well rectangular tissue culture dish (each well holds one slide), under incandescent light, for 30 minutes. Lamps with 60-Watt incandescent bulbs were positioned so the bulb was four inches above the 4-well dish. Placing slides in the outer two wells and leaving the center wells empty resulted in temperature increasing from 23 to 39 °C over 30 minutes (see Extended Data Figure 5c).”

3. Page 6 line 145: “Bleed-through was evident for weakly staining and strongly staining antibodies...” The example you provided (PD1-AF647 and CK19-AF750) is quite intriguing. Because the AF750 and AF647 should be spectrally separately using optimized filter sets, there might be a possibility that the chemical and physical factors (e.g., conjugation or oxidation) could change spectral properties of these fluorophores. It might need to be tested carefully.

Comments to Reviewer: Although AF750 and AF647 are mostly spectrally separate, as you can see in figure below, there is a slight overlap in the emission spectra of AF750 (top figure, magenta line) and the Zeiss 50 filter set emission band pass (bottom figure, red shading). The tail of the AF750 emission (i.e., at around 700 nm) is picked up by the AF647 filter set, the Zeiss 50 filter set, which has a band pass from 665 to 715 nm.

Figure 1: *Bleed Through.* a. The Alexafluor (AF) 750 excitation (blue) and emission (magenta) spectra. b. The AF647 filter set 50 excitation (blue) and emission (red) spectra. c. Specifications for the Zeiss filter set 50 used for AF647. The emission band pass, from 665 – 715 nm, overlaps with the tail of the AF750 emission spectra between 700 and 715 nm.

Revisions Made: We also observed FITC to PE bleed through in the Miltenyi MACSima and minimized it using antibody panel rearrangement, see new data analysis in Extended Figure 8h-8k. The following text was added to Antibody Order Evaluation and Improvement section:

“Bleed through was not distinct to our imaging fluorophores and filter sets, as we observed FITC to PE bleed through in another multiplex imaging platform (Extended Data Fig. 8).”

- Page 4 line 94: “...reduce overall tissue autofluorescence by roughly 25%.” Data/figure to support this point?

Comments to Reviewer: In support of this point, we added a figure to quantify the reduction in autofluorescence.

Revisions Made: In the revised manuscript, we added Extended Data Figure 5e, quantifying the reduction in autofluorescence in three adjacent slides. We now mention this figure in Quenching Condition Assessment section, as reflected in the changes to the following text:

“...to reduce overall tissue autofluorescence by roughly 25% (Extended Data Fig. 5e).”

- In the Ext. Data Figure 1 Vimentin and CD44 showed lower mean intensity compared to standard IF, relative to CycIF? Any explanation?

Comments to Reviewer: We did see differences in the raw signal and background between conditions, presumably due to the imprecisions introduced by manual pipetting during the staining procedure. We hypothesize that more antibody applied results in higher mean intensity, but also higher background. Therefore, we quantified the signal-to-background ratio (SBR), which was comparable between standard IF and CycIF. The relative SBR (see methods) of standard IF

compared to CyCIF was near one (relative SBR: mean=0.96, standard error of the mean [SEM]=0.22, Fig. 1f, Extended Data Fig. 1c).

Revisions Made: To make the source of this variation clearer, we added text to Comparison to Standard IF section:

“Although the signal and background intensity varied between conditions, presumably due to the imprecisions introduced by manual pipetting during staining, the relative signal-to-background ratio (SBR, see methods) of standard IF compared to CyCIF was near one (relative SBR: mean=0.96, standard error of the mean [SEM]=0.22) and highly correlated (Pearson R=0.89, $p=1.6e-9$, Fig. 2c, Extended Data Fig. 1c, NB1-3).”

6. Missing catalog numbers and clones in Supplementary table 1

Comment to Reviewer: We apologize for this oversight; we have now added all missing clones and catalog numbers.

Revisions Made: Please see the new Supplementary Table 1 for all catalog and clone numbers of antibodies used in our studies.

7. In Figure 2j it looks like different regions for left and right panels were used. If so, this might not be a fair comparison since different regions could have different background/AF. It would be better to find the same ROIs, using adjacent sections.

Comments to Reviewer: The figure in question does is in fact not different regions. A zoomed out view of the tissue shows it is actually the same tumor nest, but the images are collected from several sections apart. Since the slides are several sections apart, the tumor morphology does not look identical, but it is the same region, as can be determined by the ROI coordinates provided here:

https://github.com/engjen/cycIF_Validation/blob/master/OrderOptimization_K154vsK175.ipynb).

Revisions Made: In order to clarify that the slides were not directly adjacent, we added text to Antibody Order Evaluation and Improvement section:

“We compared the original and optimized orders on near-adjacent slides cut from a TMA containing three tissues with positive staining for each marker.”

8. Also, in Ext Fig 6, the CD45 in R2 and R11 show similar nuclear backgrounds. Why it's different?

Comments to Reviewer: The images show a subtle difference between nuclear background in the tumor compartment. While clear in the image, the quantification across the whole tissue did not reflect the small (but important) contribution of tumor cell nuclear background. Therefore, in quantifying the signal-to-background ratio in Figure 3i, we manually selected areas of non-specific background.

Revisions Made: We clarified our method of background selection in the text now added to Antibody Order Evaluation and Improvement section:

“Signal-to-background ratio (SBR) quantification was completed by applying a threshold to each marker to find positive pixels, and manually selecting areas of non-specific background (e.g.,

tumor nests for immune markers and stromal areas for tumor markers). Our new panel order significantly increased SBR (Fig. 3i, NB2-6).”

Reviewer #2:

We greatly appreciate the reviewer’s support of our work and the positive and constructive comments on our manuscript. We have made revisions to address the reviewer’s suggestions that are outlined below in a point-by-point manner, which we believe improve the clarity and impact of the manuscript.

General Reviewer #2 comment: *This paper reports an improved approach for tissue image analysis that is particularly applicable for highly multiplexed immunofluorescence (IF) imaging of the spatial distribution of multiple protein biomarkers. Although the paper is focused on the analysis of multiplex IF performed using the multiplex imaging platform, CyCIF, the method should be generalizable to other multiplex methods.*

Overall, this is an important contribution to the tissue image analysis field and will provide significant impetus for using methods like CyCIF to perform more robust image analysis of highly multiplexed tissues. The work establishes new protocols for removing autofluorescence background signals, order of antibody application for CyCIF, along with the development of several powerful, open-source algorithms for image processing and analysis.

The paper is suitable for publication after a few minor points are addressed:

Minor Comments:

1. Line 51-53. "Although we focus on validation of a single platform, our tools for image analysis facilitate quantitative and reproducible characterization of tissues by any multiplex imaging platform." The authors should reference here a few other platforms besides CyCIF where their approach to image analysis of highly multiplexed proteins might be applicable and give further discussion how their methods can be used for these cases. Example include the recent use of photocleavable mass-tags in conjunction with MALDI-MSI (Yagnik, G. *et al.* J Am Soc Mass Spectrom 32, 977-988); Imaging mass cytometry (Giesen, C. *et al.* Nat. Methods 2014, 11, 417-422) and in the field of transcriptomic combinatorial labeling where it is now possible to image the location of thousands of RNA species (Chen, K.H. Science 215 348 (6233, aaa66090).

Comments to Reviewer: We agree that the methods you mentioned including Imaging Mass Cytometry, MALDI-IHC and other protein labeling technologies produce data suitable for analysis using our methods. As of this writing, we have worked with data from three platforms, and now provide the Jupyter notebooks we use for processing Zeiss Axioscan images, Akoya CODEX images and Miltenyi MACSima images.

Revisions Made: We have now included of Jupyter notebook for processing Akoya CODEX images and Miltenyi MACSima images, <https://gitlab.com/engje/jinxif/-/tree/master/jupyter>. In the Code Availability section, we also added text to point to these notebooks:

“The data, code and Jupyter notebooks to reproduce the analyses herein are at https://github.com/engjen/cycIF_Validation. For image processing, we developed jinxif, available through the Python Package Index, <https://pypi.org/project/jinxif/>. We demonstrate processing Zeiss Axioscan, Akoya CODEX and Miltenyi MACSima images in our pipeline example Jupyter notebooks, here: <https://gitlab.com/engje/jinxif/-/tree/master/jupyter>.”

2. Line 64: We benchmarked our CyCIF protocol against a direct IF protocol. CyCIF staining in normal and malignant breast tissue was compared to standard direct IF on adjacent tissue sections. Since this is a major goal of the study, a summary is needed later in paper which not only compares the analytical results of the two approaches but also compares the two different work flows. For example, if equivalent results can be obtained using adjacent tissue sections, what advantages are there for performing CyCIF on the same tissue section (e.g., shorter overall processing time, etc.)?

Comments to Reviewer: The advantage of CyCIF is that since all markers are on the same slide, we gain combinatorial information about cell types. For example, it takes six markers to define proliferating PD-1+ T cells (i.e., cytokeratin-, CD31-, CD45+, CD3+, PD1+, Ki67+), which are biologically important because they represent antigen experienced T cells that may recognize the tumor. With CyCIF, many more cell types can be defined, and measured in direct relation to each other, both in number and proximity.

Revisions Made: We added the following text to Comparison to Standard IF section to clarify this point.

“We concluded that the SBR and specificity of antibody staining in the first five cycles of CyCIF is similar to standard IF, while CyCIF offers the advantage of detecting increased marker combinations while utilizing a single slide.”

3. Line 93: "to reduce overall tissue autofluorescence by roughly 25%." Is this reduction in autofluorescence true for all of the tissue types analyzed? Can the authors give more specific information (e.g., did it work for most of the different tissues studied?)

Comments to Reviewer: As described in our response to Reviewer 1, point #2, we collected two additional datasets to evaluate AF during CyCIF, totaling 246 additional tissues and extended the experiment to a full 10 rounds.

Revisions Made: Additional data was collected, where the analysis is shown in Figure 3d-3j; Extended Data Figure 10, 11 and 12. Tissue specific differences in autofluorescence are shown in Figure 4 j and Extended Data Figure 10 h.

The revised text can be seen in the response to reviewer 1, point 2 under main comments.

4. Line 118. We implement both the “scaled” and “baseline” algorithms for autofluorescence subtraction in mplex-image. Can the authors provide more guidance on when to implement scaled vs. baseline algorithm for autofluorescence subtraction?

Comments to Reviewer: After collecting AF data from 246 additional tissues, we provided guidance indicating that the scaled algorithm was a special case that could be implemented for short CyCIF experiments. The scaled algorithm only works when autofluorescence is linearly decaying or increasing, which is true up until round three or four, after which we find autofluorescence increases again. The scaled algorithm would be appropriate up until round 3 or 4, but for most tissues and experiments, we cannot perfectly subtract all autofluorescence since it tends to change round to round. Instead, we suggest subtracting the round with the lowest intensity to avoid any over-subtraction. We show that this method still removes 60-70% of autofluorescence.

Revisions Made: We updated the Autofluorescence Dynamics and Corrections section to reflect our new findings and recommendations regarding AF subtraction:

“Therefore, to computationally remove autofluorescence without over-subtracting, we recommend collecting “baseline” autofluorescence images after quenching at rounds 3 or 4, and subtracting these, scaled by exposure time, from AF488, AF555 and AF647 channels. Subtraction from the AF750 channel appears unnecessary, given its minimal autofluorescence. This procedure will remove an additional 60 – 70% of autofluorescence from the brightest rounds (Fig. 4i, Extended Data Fig. 10, 11). Since different tissues and other multiplex imaging platforms may exhibit different autofluorescence dynamics, we implement both the “baseline” algorithm and a “scaled” algorithm, which assumes linear increase or decrease in autofluorescence, and for autofluorescence subtraction in jinxif. We found that the scaled algorithm, by linearly interpolating autofluorescence between two background images, did reduce false positives for some markers in our panel (Extended Data Fig. 12). Since the brightest areas of autofluorescence show the linear decrease across the first few rounds of CyCIF, the scaled algorithm is most advantageous in tissues with strong autofluorescence that are stained only a few rounds. Conversely, the “baseline” algorithm is simple, requiring collection of just one background image, and can remove the majority of autofluorescence background without over-subtraction in various tissue and experimental contexts.”

5. Line 129-132: Non-specific nuclear staining was observed for several antibodies conjugated to the AF750 fluorophore, and unexpectedly, it diminished on the second application (Fig. 2j). Can the authors provide some suggestion as to why this might have occurred?

Comments to Reviewer: We tested several variables impacting the round-effect of stain quality including non-specific IgG interactions, non-specific fluorophore interactions, specific IgG interactions, and quenching effects. Out of all of these, the only one that apparently reduced non-specific nuclear background was additional quenching. We hypothesize that the alkaline oxidation quenching step disrupts antigen integrity. To support this hypothesis, we observed that CD45-AF750 applied to tissue that had been previously quenched 10 times had lower CD45 specific signal, but also lower non-specific binding in the nucleus.

Revisions Made: To support this point, we added Extended Data Figure 8a-8g. We updated the Antibody Order Evaluation and Improvement section to reflect our new data and findings regarding non-specific staining.

“We tested several potential variables impacting stain quality in later rounds, including non-specific IgG interactions, non-specific fluorophore interactions, specific IgG interactions, and quenching effects. Out of all of these, the only one that apparently reduced non-specific nuclear background was additional quenching (Extended Data Fig. 8).”

6. Figures: Define in legend the scale (e.g., scale bar needed) for the tissue images shown throughout (e.g., Figure 1f, 2a, 2e, 3b)

Comments to Reviewer: We agree with the reviewer’s point and have added scale bars to the tissue images throughout our revised manuscript.

Revisions Made: Scale bars were added to figures 2, 3, and 5.

Reviewer #3:

We greatly appreciate the reviewer's support of our work and the positive and constructive comments on our manuscript. We have made revisions to address the reviewer's suggestions that are outlined below in a point-by-point manner, which we believe improve the clarity and impact of the manuscript.

General Reviewer #3 comment: *Eng et al. in this paper address an actual problem, the need to evaluate through precise and shared image analysis methods the performances of multiplexing methods in order to optimize the method itself and produce results that are the real reflection of the biological processes investigated. With this work they share, as an open source, their python code (mplex-image) to offer a fully-reproducible image analysis pipeline that should allow the user to quantitatively assess antibody labeling and evaluate strategies for signal removal, antibody specificity assessment, background correction and batch normalization. They develop and apply this method using the CyCIF multiplex imaging method, offering optimization and validation to the method itself. It is true that their work has the potential to raise awareness of the need for validation studies for multiplexing methods (that too often claim performances that raise expectations that are not met in the laboratory practice). Also, offering the tools as an open source is certainly a big plus. Nevertheless, this paper doesn't meet its full potential because there are several points that could be improved.*

A first very general comment is that the authors present a missing need in the literature related to multiplexed IF analysis. Most of the analyses are performed in only one type of tissue/cancer, and we have concerns about how generalizable their results are. In our experience, multiplexed images tend to be very heterogeneous (tissue type, pathology, Abs used, etc.) so what is true for pancreatic tissue might not be true for breast tissue.

General Comments to Reviewer: We appreciate the reviewer's point about tissue-to-tissue differences, and we have now increased the types of tissue analyzed to address this point. Specifically, we analyzed autofluorescence and tissue loss in three 72-core tissue microarrays containing diverse normal and malignant tissues. We believe these analyses will be helpful to those trying to anticipate tissue loss or autofluorescence in various tissue and tumor types. Additionally, the 51 antibodies we report on in this work cover epithelial, immune and stromal targets that are expressed in a wide variety of tissue types, not only breast (including dozens of different tumor and cell line specimens), tonsil and pancreas that are specifically stained in this paper.

Specific Comments:

1. The authors use their pipelines to validate and optimize CycIF. Line 37 and Line 52: Very limited literature about iterative fluorescence methods, a lot of them are not listed here. For completeness' sake, the authors should research literature and list all the IF-based cyclic methods: do the authors foresee differences applying their framework to the other methods? How the pipeline may need to be adjusted?

Comments to Reviewer: As recommended, we have expanded our literature review (please see below). As mentioned in our response to reviewer 1, we also added two Jupyter notebooks showing processing of images from the Akoya CODEX and Miltenyi MACSima Instruments. For these platforms, we start by standardizing image metadata, and then enter the images into our pipeline at the cell segmentation step. After the image metadata is standardized, we only need to adjust parameters for cell size, due to the fact that different platforms have different image sizes.

Revisions Made: Please see our response to Reviewer 1, point 1 (main comments) for revisions related to adjusting the pipeline for other methods. The main text was updated in Introduction section to review existing multiplex imaging methods, as shown below.

“Five and seven-plex immunohistochemistry (IHC) can be achieved by fluorophore-conjugated tyramide deposited on the tissue¹⁻³. Twelve to 29-plex IHC is enabled with alcohol-soluble peroxidase substrate 3-amino-9-ethylcarbazole (AEC) detection combined with antibody stipping^{4,5} and 40-plex IF can be achieved with antibody elution in iterative indirect immunofluorescence imaging (4i)⁶. Direct immunofluorescence using fluorophore-conjugated primary antibodies and chemical inactivation of fluorescent dyes enables detection of over 50 protein targets in a single tissue section, in cyclic immunofluorescence (CyCIF)⁷⁻⁹ and NeoGenomics’ MultiOmyx platform¹⁰. Furthermore, fluorophore-conjugated DNA oligonucleotides facilitate multiplexing in co-detection by indexing (Akoya’s CODEX)¹¹, Ultivue’s InSituPlex¹², and Ab-oligo CyCIF¹³. Imaging mass cytometry¹⁴ and multiplex ion beam imaging¹⁵ can detect over 40 antibodies conjugated to metal reporters by time-of-flight mass-spectrometry. The Nanostring GeoMx¹⁶ platform can detect over 100 protein targets conjugated to oligonucleotide barcodes, although the data are not images, but spatially registered counts of released oligos¹⁷.”

2. The authors do benchmarking with standard IF (visual inspection + SBR + positive cells count) and they conclude that the SBR and specificity of antibody staining in the first five cycles of CyCIF is similar to standard IF. Moreover, they detect tissue loss and point to the need for gentler quenching, identifying quenching using 3% H₂O₂ for 30 minutes with incandescent light + an additional round of pre-quenching as the best modification for the method (though the pre-quenching is not novel, as they admit). Line 66: how could the authors then validate markers that are negative in the section of interest? Do they expect to always have a positive control inside the reference tissue? Were these negative markers discharged by default?

3. **Comments to Reviewer:** In order to validate all antibodies, we had to use two control tissues, HER2+ breast cancer and normal breast. Between the two tissues, we had at least one positive control for each of our markers. When one of the tissues was negative for a marker of interest, we did not include it in the analysis. Therefore, we validate 15 antibodies in two tissues, but our final n for standard vs CyCIF was 26 (Figure 2c). This is due to dropping markers that were negative in a given tissue (e.g., HER2 and PD-1 in normal breast and CK14 and CK5 in luminal HER2+ tumor).

Revisions Made: We updated text in Comparison to Standard IF section to clarify the reviewer’s point about tissues negative for particular markers.

“If a marker was negative in a given tissue section, it was not analyzed in that tissue.”

4. Line 67: the authors talk about “quantification of the signal” using a threshold, but this is not quantification of the signal, it’s a count of cells considered as “positive” without taking into consideration the full range of value of the signal. Moreover, thresholds were set manually in small ROIs. I have concerns about how generalizable these thresholds are to other regions of the images.

Comments to Reviewer: This is a valid point which we have addressed by changing the wording of our explanation (see below). Once we set an intensity threshold, we can calculate (1) number of positive cells, (2) the mean intensity of positive pixels and (3) background, (4) the signal-to-background ratio (SBR), and (5) the dynamic range, resulting in a variety of calculated values using the threshold. To evaluate the generalizability of thresholds to other regions of the image, we selected new ROIs, one in the set of five normal breast slides and another in the set of five tumor slides. We applied the same thresholds to both original and new ROIs and calculated the correlation between ROIs for mean signal intensity and background for each marker. We found high correlation between ROIs, as shown in Extended Data Figure 2.

Revisions Made: Our quantifications in figure 2c show percent positive cells and SBR. Extended Data Figure 1c shows signal intensity, background intensity and dynamic range. We added Extended Data Figure 2 to investigate the Regional Variation in Threshold Results. We changed the wording in Comparison to Standard IF section to remove any potential confusion around the wording “quantification of signal.”

“For quantification, we set an intensity threshold for each marker, with all pixels above the threshold considered positive. All pixels at least 10 micrometers away from positive staining were considered background pixels (the 10 μm gap was to exclude the influence of lateral bleed around positive pixels).”

We also add discussion of thresholding in the methods:

“We tested whether the same threshold could be applied to different regions of the same slide by measuring correlation between two ROIs in normal breast and two ROIs in HER2+ breast tumor given the same threshold. The mean fluorescence intensity measured above threshold, and the intensity of background noise were highly correlated between ROIs (Extended Data Figure 2).”

5. Line 68 (NB1-2, https://github.com/engjen/cycIF_Validation/blob/master/Extended_single_vs_cyclic.ipynb): the authors show various images heavily affected by the field-of-view-artifact. However, they do not mention any flat-field-correction for their images throughout the manuscript. This can affect the results obtained by antibodies with low SNR.

Comments to Reviewer: It is possible to make flat-field corrections using the Zeiss Zen software, however, most of the antibodies utilized here for staining resulted in $\text{SBR} > 1.5$ so flat field correction was not required due to sufficient signal to noise ratio. See Extended Data Figure 7 for quantification of SBR.

Revisions Made: We updated the methods section to discuss flat field correction.

“For the datasets used in this work, we did not apply flat field correction, although it may be applied in the standard Zeiss Zen software using the “Shading Correction” function.

6. Line 59 and 75: the authors state that 40-60 protein could be labeled (10-15 cycles) before tissue deterioration, but also that SBR and antibody specificity is similar in CyCIF and standard IF for 5 cycles: how can these two sentences coexist? How valuable can the comparison with standard IF be for markers that are stained in later cycles?

Comments to Reviewer: Although we only compare standard IF to 5-round CyCIF, we established that the cyclic process is not altering the tissue or staining compared to standard IF and our analytical results are similar. Therefore, if we see expected staining patterns and sufficient SBR after 5 rounds, we can reasonably assume that the CyCIF result is comparable to a standard IF result. In Extended Data Figure 7: Quantification of first versus second antibody application to the same TMA tissue calculate SBR in 22-round CyCIF. Our analysis finds 23 of 37 antibodies with SBR > 1.5 in rounds 12-22. Placing antibodies that result in robust staining to cycle number in these later rounds enables 15+ cycles. While later round cycles do lose some dynamic range compared to earlier rounds, we show they are still analyzable.

Revisions Made: The following text was added to Results section to reference the data regarding 40 – 60 proteins per slide.

“We typically labeled 40 – 60 proteins per slide before tissue and staining quality degraded (quantified in Figure 2e and Extended Data Figure 7, respectively).”

And we added an additional concluding statement about the cyclic process in general, to our Comparison to Standard IF section:

“We concluded that the SBR and specificity of antibody staining in the first five cycles of CyCIF is similar to standard IF and the cyclic process does not excessively impact tissue staining.”

7. Line 78: the authors talk about “increased tissue loss,” do they specifically mean physical detachment of the tissue from the slide? Loss of antigenicity of the tissue? Or both? From the images in Extended Data Fig. 1a we can assume both. The legends of the images are not sufficiently detailed: what are we supposed to see from the images that they provide? There is always written what the graph/image is about, but not a description of what it is meant to show.

Comment to Reviewer: By tissue loss, we mean physical detachment of the tissue from the slide. We have now added analysis of tissue loss to Figure 2d – 2f and Extended Data Figure 3. We also added detail to the figure legends to make it clearer that we are trying to show tissue loss in Extended Data Figure 1.

Revisions Made: We added arrowheads to point out areas of tissue loss (i.e., detachment from the slide) in Extended Data Figure 1. Additionally, we analyzed 216 more tissues for 10 rounds to test variables effecting tissue retention during CyCIF, see figure 2d-2h and Extended Data Figure 3. We also added clarifying text to the figure legend in Extended Data Figure 1.

“Tissue overview, display range in x-axis label. Round (R) 1 on left, versus R4 on right. Arrowheads show areas of tissue loss.”

8. Subsequently, the authors attempt an original approach on autofluorescence removal: they investigate single-cell autofluorescence (AF) changes and highlight that one of the AF components following linear changes - based on this the authors implemented a “scaled” next to a “baseline” algorithm for autofluorescence subtraction in mplex-image. Line 102: “Overlaying the cells in each cluster on the tissue revealed cells in cluster 7 comprised tissue structures with bright autofluorescence, such as elastic fibers” → elastic fibers should not be nucleated structures. This

evidence makes the segmentation approach used in this paper not believable since it includes measuring of the background rather than of the real cytoplasm of the cells. Moreover, in figure 2e it is not clear in showing why the authors are sure that those structures are elastic fibers - the images are too small and not well readable.

Comments to Reviewer: We understand the confusion and took out reference to elastic fibers. Although our pathology collaborators noted that these are apparently elastic fibers surrounding a blood vessel in the normal pancreas, they have been segmented because they also appeared in the DAPI channel. We believe this is an example of AF488 to DAPI bleed through that could be mitigated by pre-quenching. For subsequently analyzed datasets (72-core TMA and 11 core TMA) we segmented on DAPI staining after two rounds of quenching.

Revisions Made: The main text was updated in Autofluorescence Dynamics and Corrections section to remove “elastic fibers” and discuss the hypothesized bleed through:

“Overlaying the cells in each cluster on the tissue revealed cells in cluster 7 comprised tissue structures with bright autofluorescence, that were bright enough to exhibit bleed through from the AF488 to DAPI channels”

9. Line 106-108: based on the results shown in figure 2g, the assumption that AF decays linearly for other components seems to be oversimplifying. Image intensities are affected by a large series of variables including: noise caused by imperfections in image acquisition (see: <https://ieeexplore.ieee.org/document/5590292>). This might affect markers with low SNR where true positives are slightly above background. Line 113: the authors give the definition of what they consider as false positive based on measuring AF at single cell level, but what do they consider as “false negative?” No definition is given.

Comments to Reviewer: Thank you for pointing out that the linear AF decay may be oversimplifying. On further analysis of 246 additional tissues, we found more complex pattern of AF falling and rising over 10 rounds of CyCIF. See our response to reviewer 1, point 2 in main comments for our discussion of these findings. To your other point, when we compare the baseline and scaled algorithms, we do a limited evaluation of true positives, false negatives and false positives to estimate the F1 score (true positives)/(true pos + 0.5*(false pos. + false neg.)) of each method. The analysis was limited because it required manual annotation of true positives and false positives using code we developed for annotation in the napari image viewer.

Revisions Made: In Extended Data Figure 8: Single Cell Annotation for Evaluation of Autofluorescence Subtraction, we add a description of false negative (see b): Example of CD8 positive/negative annotation in napari image viewer (+ and – overlaid on cell segmentation, center, and CD8 staining, right). The following text was added to methods section for further clarification:

“For F1 score calculation in Extended Data Figure 11 we again used the napari image viewer to overlay staining, segmentation results, and positive cells based on manual thresholding. Based on the staining pattern and other marker’s expression (e.g., membranous CD8 staining in CD45+ cells was a true CD8 positive), we manually annotated false negatives in three 2000 x 2000 pixel ROIs. False positives were any cell with AF488 autofluorescence > 1024, true positives were cells above

threshold excluding false positives and true negatives were all other cells neither positive, false positive or false negative.”

10. A very nice paragraph in the paper is where the authors measures sensitivity and specificity of the antibodies in relation to the round in which they are performed (early vs late) in order to optimize the panel design following the results. This is definitely a fundamental concept for all cyclic methods. Line 122: regarding Supplementary table 2, the rounds considered as “early” are until round 10 - yet, since at the beginning it was stated that the images are comparable to the gold standard until round 5, we wonder if it was a good strategy to include 10 rounds in the definition of “early” rather than limit the validation to less antibodies stained twice in a total of 10 rounds in total (5 early and 5 late).

Comments to Reviewer: Thank you for pointing out the potential confusion around the use of the term “early.” We altered our terminology to the more precise “first application” and “second application” rather than “early” and “late” in the revised manuscript.

Revisions Made: We updated our terminology in Extended Data Figure 6: First versus second antibody application to same TMA tissue. We edited the text in Extended Data Figure 6 legend to reflect these terminology changes.

“First versus second antibody application to same TMA tissue. a. CK19 in R1 (first) and again in R12 (second) shows decrease in dynamic range. b. CD4 shows little decrease in dynamic range. c. CD45 shows improvement in non-specific nuclear staining on second application. d. PD1-AF647 has clear bleed through from CK19-AF750 (matching pattern in a), in both applications. f. Single cell mean intensity distribution of positive signal (S) or background (BG). Cell types were defined by thresholding and gating. (Signal = positive cells for respective marker. Background = Tumor cells for stromal markers, and vice versa. Additional markers’ visualizations and figures here: https://github.com/engjen/cycIF_Validation).”

11. Finally, the authors perform a reproducibility test: they stain three consecutive sections of a TMA on three different dates and compare two methods to evaluate the dynamic range of the stainings. With both they obtain similar results, highlighting reproducibility of the method but also the need for batch effect correction. They then compare three batch effect methods. All improved the clustering. Combat was found to be better, even though it needs the presence of control tissue, while RESTORE works better with mutually exclusive markers and does not need control tissue. This was done on tissue sections and repeated on cell line to check cell type identification. Some more remarks regarding the methods: Line 324: “Nuclear segmentation accuracy was improved by sorting nuclei based on expression of tumor cytokeratins or immune markers.” This sentence is not clear – cytokeratins are cytoplasmic staining, how can a cytoplasmic staining improve the segmentation of the nuclei? Please explain.

Comments to Reviewer: Since epithelial cells have larger nuclei than non-epithelial cells, the segmentation algorithm uses the cytokeratin staining to find epithelial cells and then uses different parameters for those nuclei (i.e., allows them to be larger). This is only for the watershed segmentation, which tends to “overflow” and make nuclei too large if we do not specify a maximum nuclear size. Since our tumor tissues had variable nuclear size, from 10 μm in leukocytes to 30 μm in tumor cells, we found it necessary to sort the nuclei when using watershed segmentation. The deep learning segmentation algorithm Cellpose does not need cytokeratin to sort nuclei and requires only DAPI for segmentation. Cellpose does require a nuclear size

parameter, which we set at 30 pixels, but the algorithm was able to accurately segment all nuclear sizes present in our tissues. We tested different nuclear diameter values at <http://www.cellpose.org/> to determine the best one for our images.

Revisions Made: Text was added to methods to clarify how cytokeratin staining was used in segmentation.

“Nuclear segmentation accuracy was improved by sorting nuclei based on expression of tumor cytokeratins or immune markers and using this information to set a maximum nuclear size for the watershed algorithm. If a cell was positive for cytokeratins, it was allowed to have a larger nucleus than cells that were negative for cytokeratins.”

12. Line 326 – cell segmentation: how is the 5 pixel expansion justified? Why 5 pixels and not 2 or 10?

Comments to Reviewer: We measured the immune cell cytoplasm (i.e., the distance from DAPI to CD45 staining) and found it to be roughly 5 pixels, which corresponds to 1.625 μm given our pixel size of 0.325 μm . Since our deep learning-based segmentation used E-cadherin (E-cad) to segment epithelial cell membranes, this left E-cad negative immune and stromal cells without a segmented membrane. Therefore we used the estimated immune cell cytoplasm of 5 pixels to expand the E-cad negative nuclei.

Figure 2: *Measurement of immune cell cytoplasm.* a. Measurement of width of immune cell cytoplasm. Blue = DAPI, yellow = CD8. b. Result of 5 pixel expansion. Blue = DAPI, yellow = CD8, inner brown ring = nuclear segmentation, outer brown ring = cytoplasm segmentation. Cells in upper left corner are tumor cells and Ecad was used to segment cell boundary. Cell on right are immune cells and 5 pixel expansion was used to segment cell boundary.

Revisions Made: Text added to methods to clarify why 5 pixels was selected.

“Nuclei with no E-cadherin staining (i.e., non-epithelial cells) were expanded by 5 pixels (1.6 micrometers) to approximate the cytoplasm, based on the average measurement of immune cell cytoplasm in representative images.”

13. Line 327: “membrane” written here gives the false impression that the authors evaluate also the membrane location like a third subcellular area, while later it becomes clear they subsegment in nuclei and cytoplasm.

Comments to Reviewer: We used two segmentation algorithms for these data sets. For the python-based algorithm, we can segment the membrane, since the deep learning segmentation used E-cadherin to predict epithelial cell membranes. However, we did not use the membrane segmentation for the datasets analyzed in this work, so we reworded our methods.

Revisions Made: We removed the word “membrane” in the methods section on segmentation, instead referring only to the cytoplasm.

“The cytoplasm was derived by subtracting the nuclei area from the cell segmentation result, or from the 5-pixel expansion result in the case of Ecad negative cells.”

14. Line 342: “all cells with a mean intensity above the set threshold were considered positive.” How were these thresholds calculated? How was the decision made?

Comments to Reviewer: Please see our response to reviewer 1, point 4 under major comments.

Revisions made: The text added to methods is detailed in this response to reviewer 1.

15. Line 344: Not all the cells have the same background/foreground and therefore not the same SBR. Moreover, since positive/negative pixels are selected based on manual thresholds, they have control on the estimation of the SBR.

Comments to Reviewer: Thank you for pointing out the confusion around single cell analysis and integration across the slide. It is true that not all cells have the same SBR, our analysis in that figure was more concerned with comparing mean SBR between different conditions or replicates than single cell SBR. We have now clarified the methods. Moreover, we understand the problem with using manual thresholds but we do not yet have a good method of automating thresholding. Therefore, we used manual thresholds but did not select the thresholds based on SBR, but rather on the pixel patterns (i.e., we sought to select equivalent pixels in each image for comparison). See response to reviewer 1, point 4 under major comments for discussion.

Revisions Made: We clarified the SBR methods to distinguish single cell analyses from integration across the entire tissue. The following text was updated in methods section to reflect these changes.

“For single cell analysis, single cell mean intensity was used for clustering, as in Figure 4 and 5. For percent positive calculation, as in Figure 2c (left), cells with a mean intensity above the threshold were considered positive. Tissue retention was calculated in Figure 2e – 2h by thresholding DAPI using the Li algorithm²⁷ and considering cells above DAPI threshold as retained in that round. For signal-to-background calculations, mean intensity was integrated across the entire slide or region of interest, as in Figure 2c, right, Figure 3, and Figure 5b.”

16. Line 348: How does the estimation of the dynamic range in the q5-q95 range affect rare markers expressed in very few cells (~0.1%)?

Comments to Reviewer: To address this point, we tested different quantile values for estimating dynamic range and adjusted our dynamic range calculation to q4 to q99.5 to better reflect the dynamic range of rare markers.

Revisions Made: See Extended Data Fig. 14 for comparison of dynamic range estimations. We added text to the methods to reflect these new data.

“Dynamic range was estimated using the 4th and 99.5th quantile of mean intensity for markers in tissues with known positive staining. We compared different ranges for estimating dynamic range (5th to 98th percentile versus 5th to 99.9 percentile, see Extended Data Figure 14). We found that using a higher maximum did not change the dynamic range as much for common markers (e.g., CK7) but had more effect on rare markers (e.g., Ki67 and alpha-SMA). Therefore we selected the 99.5th percentile as the maximum to reflect both common and rare markers’ dynamic range.”

REVIEWERS' COMMENTS:

Reviewer #1 (Remarks to the Author):

The revised manuscript has fully addressed my concerns raised previously. The additional data and information in the current version significantly strengthen the points authors would like to deliver. Thus, I recommend to publish this on Communications Biology.

A few minor comments below:

(1). In figure 2c, the authors demonstrated the correlation between standard IF and CyCIF using number positivity & Pearson correlation. However, this might be somehow misleading given most of the data points with very low positive cells/pixels. The log version of the same analysis or other type of statistical tests might be more suitable here.

(2). I'm excited to see the authors' attempt to address the issue of cell/tissue loss. However, the use of multi-tissue TMA could be somehow problematic. Since all the tissues might not be processed with the same protocol prior reconstructing into TMA, these technical variables could theoretically contribute to the tissue loss, while the contributions of tissue types are hard to determine. I might suggest the authors to at least mentioned these limitations in the final version.

(3). I'm glad to see the improvements made in the antibody panel design (figure 3 & related text). However, it might be even more informative if the authors provide a schematics or flowchart to help readers understand the principles.

(4). In the extended Figure 8, the CK7 (luminal marker) and CK14 (basal marker) are totally overlapped, why?

(5). Line 182-183 "...bright enough to exhibit bleed through from the AF488 to DAPI channels..". Given the autofluorescence could cover wide spectrum including DAPI, the term "bleed through" might not be appropriated here.

(6). About the backward bleed through between AF750 and AF647, I did appreciate your explanation in the rebuttal. However, based the vendor information (attached) as well as the filters you provide, I still have to imagine this phenomenon is simply due to the channel split-over. If so, you might want to check with your microscope provider, as well as conducting some control experiments with standard IF to address this in the future.

Reviewer #3 (Remarks to the Author):

We wish to thank the authors for the amazing job done with addressing our comments. In our opinion, only the references to different fluorescence-based multiplex methods are incomplete (while imaging via mass spectrometry was unexplainedly included). Please add the following references as well:

Schubert, W. et al. Analyzing proteome topology and function by automated multidimensional fluorescence microscopy. *Nat. Biotechnol.* 24, 1270–1278 (2006).

Bolognesi, M. M. et al. Multiplex Staining by Sequential Immunostaining and Antibody Removal on Routine Tissue Sections. *J. Histochem. Cytochem.* 65, 431–444 (2017).

Saka, S. K. et al. Immuno-SABER enables highly multiplexed and amplified protein imaging in tissues. *Nat. Biotechnol.* 37, 1080–1090 (2019).

Cattoretto, G., Bosisio, F. M., Marcellis, L. & Bolognesi, M. M. Multiple Interactive Labeling by Antibody Neodeposition (MILAN). (2018).

Radtke, A. J. et al. IBEX-A versatile multi-plex optical imaging approach for deep phenotyping and spatial 2 analysis of cells in complex tissues 3 High dimensional imaging, immune system, quantitative microscopy, tissue immunity. *bioRxiv* (2021). doi:10.1101/2020.11.20.390690

Pascual-Reguant, A. et al. Multiplexed histology analyses for the phenotypic and spatial

characterization of human innate lymphoid cells. *Nat. Commun.* 2021 12:1–15 (2021).
With this last addition, we can state to be completely satisfied with the revision.

Knight Cancer Institute
Center for Spatial Systems Biomedicine
Department of Biomedical Engineering
Oregon Health & Science University

tel (503) 494-6500
cell (503) 708-7767
grayjo@ohsu.edu
<http://www.ohsu.edu/ocssb>

2730 S.W. Moody Ave., CL3G
Portland, OR 97201-5042

February 17, 2022

Eve Rogers, Ph.D.
Associate Editor
Communications Biology
4 Crinan Street
London N1 9XW, UK

Dear Reviewers,

Thank you for your support of our revised manuscript entitled “*A framework for multiplex imaging optimization and reproducible analysis.*” We have addressed your additional questions and concerns, and are pleased to provide our updated and improved manuscript as well as the following point-by-point revisions to address the specific reviewers’ comments. These changes are also shown in red text in the revised manuscript to aid in rereview as needed.

Reviewer #1:

We would like to thank the reviewer for the helpful and constructive comments on our manuscript. We have made revisions to address the reviewer’s suggestions that are outlined below in a point-by-point manner.

General Reviewer #1 comment: *The revised manuscript has fully addressed my concerns raised previously. The additional data and information in the current version significantly strengthen the points the authors would like to deliver. Thus, I recommend to publish this in Communications Biology.*

General Comments to Reviewer: We appreciate the reviewer’s support and thank the reviewer for the helpful feedback that led to the current version.

Minor Comments:

1. In figure 2c, the authors demonstrated the correlation between standard IF and CyCIF using number positivity and Pearson correlation. However, this might be somewhat misleading given most of the data points have very low positive cells/pixels. The log version of the same analysis or other type of statistical tests might be more suitable here.

Comments to Reviewer: This is a good point as the number positive depends on the number of cells sampled and we had different sized tissues in the normal and tumor tissue sections. Therefore, we revised our analysis to normalize by the number of cells per section and now display fraction of positive cells for each marker/tissue (please see Fig. R1 below). The Pearson correlation assumptions are now better satisfied because the normalization by tissue size leads to fewer outliers.

Figure R1. Number Positive versus Fraction Positive. a. Histogram of number positive cells per marker, tissue shows outliers. b. Histogram of positive cells normalized by number of cells in tissue (i.e., fraction positive) shows fewer outliers.

Revisions Made: In the revised manuscript, we updated figure 2c, which now displays the data as fraction positive cells. The following text was edited in the manuscript:

“the fraction positive was highly correlated (Pearson $R=0.99$, $p=5.8e-24$, Fig. 2c, Extended Data Fig. 1c, NB2-2).”

- I'm excited to see the authors' attempt to address the issue of cell/tissue loss. However, the use of multi-tissue TMA could be somehow problematic. Since all the tissues might not be processed with the same protocol prior reconstructing into TMA, these technical variables could theoretically contribute to the tissue loss, while the contributions of tissue types are hard to determine. I might suggest the authors to at least mentioned these limitations in the final version.

Comments to Reviewer: This is a good point about technical variables. The source of most tissues was autopsy; however, some of the normal tissues were obtained from surgical resection. We compared the normal tissue source versus tissue loss and found a trend towards better tissue/cell retention in the surgical resections vs. autopsy samples ($p = 0.083$, $n=12$ surgical resections, $n=29$ autopsy).

Revisions Made: We added a panel for “source” to Fig. 2f and the following text to the manuscript:

“While all malignant tissues were obtained from autopsy, normal tissues obtained from surgical resections vs. autopsy showed a trend towards reduced tissue loss (Mann-Whitney U, $p=0.083$). This could reflect tissue processing variation, as smaller tissue size and longer time in fixation have been shown to reduce section detachment in immunohistochemistry¹⁹.”

- I'm glad to see the improvements made in the antibody panel design (figure 3 & related text). However, it might be even more informative if the authors provide a schematics or flowchart to help readers understand the principles.

Comments to Reviewer: This is a nice idea, which we have implemented into the revised figure 3.

Revisions Made: We added cartoons in Fig. 3j to illustrate panel optimizations and updated the figure caption appropriately.

4. In the Extended Figure 8, the CK7 (luminal marker) and CK14 (basal marker) are totally overlapped, why?

Comments to Reviewer: Extended Figure 8h is an example of CK7-FITC channel crosstalk (bleed through) into Cytokeratin-14-PE, so this is actually not overlap of a luminal and basal marker, but an example of a bleed through artifact. In the improved panel order (Ext. Fig. 8 i and j) we put CK7 and Calponin in adjacent channels as well as β -Tubulin and CK14 in adjacent channels to minimize any bleed through.

Revisions Made: We added an arrow to the figure to indicate the artifact and updated the figure legend text to read:

“Bleed through example in round 1 (R1), from bright Cytokeratin-7-FITC to dim Cytokeratin-14-PE, bleed through shown with arrowhead”

5. Line 182-183 "...bright enough to exhibit bleed through from the AF488 to DAPI channels..". Given the autofluorescence could cover wide spectrum including DAPI, the term "bleed through" might not be appropriated here.

Comments to Reviewer: We agree that the terminology should be changed and have updated the text.

Revisions Made: The text was updated to read:

“bright autofluorescence that covered a wide spectrum from DAPI to AF488 channels (Fig. 4a)”

6. About the backward bleed through between AF750 and AF647, I did appreciate your explanation in the rebuttal. However, based the vendor information (attached) as well as the filters you provide, I still have to imagine this phenomenon is simply due to the channel split-over. If so, you might want to check with your microscope provider, as well as conducting some control experiments with standard IF to address this in the future.

Comments to Reviewer: We thank the reviewer for this point and plan to conduct additional control experiments in the future to fully understand this phenomenon.

Reviewer #3:

We greatly appreciate the reviewer’s support of our work. We have made revisions to address the reviewer’s suggestions for additional references.

General Reviewer #3 comment: *We wish to thank the authors for the amazing job done with addressing our comments. In our opinion, only the references to different fluorescence-based multiplex methods are incomplete (while imaging via mass spectrometry was unexplainedly included). Please add the following references as well:*

1. Schubert, W. et al. Analyzing proteome topology and function by automated multidimensional fluorescence microscopy. *Nat. Biotechnol.* 24, 1270–1278 (2006).
2. Bolognesi, M. M. et al. Multiplex Staining by Sequential Immunostaining and Antibody Removal on Routine Tissue Sections. *J. Histochem. Cytochem.* 65, 431–444 (2017).

3. Saka, S. K. et al. *Immuno-SABER enables highly multiplexed and amplified protein imaging in tissues*. *Nat. Biotechnol.* 37, 1080–1090 (2019).
4. Cattoretti, G., Bosisio, F. M., Marcellis, L. & Bolognesi, M. M. *Multiple Interactive Labeling by Antibody Neodeposition (MILAN)*. (2018).
5. Radtke, A. J. et al. *IBEX-A versatile multi-plex optical imaging approach for deep phenotyping and spatial 2 analysis of cells in complex tissues 3 High dimensional imaging, immune system, quantitative microscopy, tissue immunity*. *bioRxiv* (2021). doi:10.1101/2020.11.20.390690
6. Pascual-Reguant, A. et al. *Multiplexed histology analyses for the phenotypic and spatial characterization of human innate lymphoid cells*. *Nat. Commun.* 2021 121 12, 1–15 (2021).

With this last addition, we can state to be completely satisfied with the revision.

General Comments to Reviewer: We would like to thank the reviewer for the helpful list of additional references. We have added the six suggested references.

Revisions Made: We added the following text to our introduction (with new additions highlighted in red.)

“Twelve to 29-plex immunohistochemistry (IHC) is enabled with alcohol-soluble peroxidase substrate 3-amino-9-ethylcarbazole (AEC) detection combined with antibody stripping^{4,5} and 40-plex IF can be achieved with antibody elution in iterative indirect immunofluorescence imaging (4i)⁶ and **multiple interactive labeling by antibody neodeposition (MILAN)**^{7,8}. Direct immunofluorescence using fluorophore-conjugated primary antibodies and chemical inactivation of fluorescent dyes enables detection of over 50 protein targets in a single tissue section, in cyclic immunofluorescence (CyCIF)^{9–11}, NeoGenomics’ MultiOmyx platform¹², and **iterative bleaching extends multiplexity (IBEX)**¹³. Similarly, **multi-epitope-ligand cartography (MELC), employs photo-inactivation of fluorescently labeled antibodies**^{14,15}. Furthermore, fluorophore-conjugated DNA barcodes (i.e., oligonucleotides) facilitate multiplexing in co-detection by indexing (Akoya’s CODEX)¹⁶, Ultivue’s InSituPlex¹⁷, **Immuno-SABER**¹⁸, and Ab-oligo cyCIF¹⁹.

We thank the reviewers for their efforts to improve our manuscript. Please feel free to reach out to us should any other questions or additional information be necessary.

Sincerely,

Joe W. Gray, Ph.D.

Gordon Moore Endowed Chair, Biomedical Engineering
Associate Director for Biophysical Oncology, OHSU Knight Cancer Institute